CERN-TH-2024-120

# WZW terms without anomalies: generalised symmetries in chiral Lagrangians

**Joe Davighi**[a] **and Nakarin Lohitsiri**[b]

[a] *Theoretical Physics Department, CERN, 1211 Geneva 23, Switzerland*

[b] *Department of Mathematical Sciences, Durham University, Upper Mountjoy, Stockton Road, Durham, DH1 3LE, United Kingdom*

*E-mail:* joseph.davighi@cern.ch, nakarin.lohitsiri@durham.ac.uk

ABSTRACT: We consider a 4d non-linear sigma model on the coset $(\mathrm{SU}(N)_L \times \mathrm{SU}(N)_R \times \mathrm{SU}(2))/(\mathrm{SU}(N)_{L+R} \times \mathrm{U}(1)) \cong \mathrm{SU}(N) \times S^2$, that features a topological Wess–Zumino–Witten (WZW) term whose curvature is $\frac{n}{24\pi^2}\mathrm{Tr}(g^{-1}dg)^3 \wedge \mathrm{Vol}_{S^2}$ where $g$ is the $\mathrm{SU}(N)$ pion field. This WZW term, unlike its familiar cousin in QCD, does not match any chiral anomaly, so its microscopic origin is not obviously QCD-like. We find that generalised symmetries provide a key to unlocking a UV completion. The $S^2$ winding number bestows the theory with a 1-form symmetry, and the WZW term intertwines this with the $\mathrm{SU}(N)^2$ flavour symmetry into a 2-group global symmetry. Like a 't Hooft anomaly, the 2-group symmetry should match between UV and IR, precluding QCD-like completions that otherwise give the right pion manifold. We instead construct a weakly-coupled UV completion that matches the 2-group symmetry, in which an abelian gauge field connects the QCD baryon number current to the winding number current of a $\mathbb{C}P^1$ model, and explicitly show how the mixed WZW term arises upon flowing to the IR. The coefficient is fixed to be the number of QCD colours and, strikingly, this matching must be 'tree-level exact' to satisfy a quantization condition. We discuss generalisations, and elucidate the more intricate generalised symmetry structure that arises upon gauging an anomaly-free subgroup of $\mathrm{SU}(N)_{L+R}$. This WZW term may even play a phenomenological role as a portal to a dark sector, that determines the relic abundance of dark matter.

# 1 Introduction

Symmetries provide powerful tools for understanding the structure and dynamics of strongly interacting quantum theories. Beginning in the 1960s, current algebra proved extremely successful in explaining the organising structure of the light pseudo-scalar mesons in quantum chromodynamics (QCD) [1, 2]. This was famously refined

after the discovery of the chiral anomaly by Adler [3], Bell and Jackiw [4], through which quantum effects were shown to modify the Ward identities one would otherwise infer using the classical Lagrangian. Moving into the 1970s, Wess and Zumino deployed this anomalous current algebra to explain the observed violation of naïve spatial parity symmetry and pion number mod 2 [5], in *e.g.* the decays of the $\phi$ meson to both $K^+K^-$ and $\pi^0\pi^+\pi^-$ final states. The anomaly was also shown to account for why the $\eta'$ meson was heavier than its apparent sibling the $\eta$ meson [6–8]. The topological Wess–Zumino–Witten (WZW) term in the action matches this anomalous current algebra relation in the IR, furnishing the (previously local) current algebra structure with a global aspect [9].

Fast forward half a century, and symmetry continues to offer rich and surprising insights into the dynamics of QFTs. A key development in the last decade has been the discovery of *generalised* notions of symmetry [10], beyond the action of groups on local operators. One example is higher $p$-form symmetries, in which abelian groups act not on local operators but on extended objects. From the viewpoint of current algebra, the corresponding symmetry currents for $p$-form symmetries (in the case of continuous symmetry) are not vectors $j_\mu^{(1)}$ (better, 1-forms) as they are for 'ordinary' 0-form symmetries, but tensors $j_{\mu\ldots\sigma}^{(p+1)}$ with more than one index (specifically, $(p+1)$-forms). Returning to QCD and its discrete symmetries, $\mathrm{SU}(n)$ gauge theory possesses such a $\mathbb{Z}_n$ valued 1-form symmetry, which was shown by Gaiotto, Kapustin, Komargodski and Seiberg to have a mixed anomaly with parity when the QCD theta angle equals $\pi$ [11]. This subtle anomaly in QCD, discovered nearly 50 years after the ABJ anomaly, means that QCD at $\theta = \pi$ cannot be trivial in the deep infrared (IR).

It has also been recently understood that there can be a non-trivial current algebra between higher-form symmetries of different degree. The simplest example is known as a 2-group symmetry structure, wherein 0-form and 1-form symmetries 'mix', as first detected in quantum field theories by Sharpe [12]. In the case where both 0-form and 1-form symmetries are continuous, this can be captured, following the tradition of current algebra, by a Ward identity of the form [13]:

$$\langle i\partial^\mu j_\mu^{(1)a}(x) j_\nu^{(1)b}(y)\rangle = \langle -\delta(x-y)f^{abc}j_\nu^{(1)c} + \frac{n}{8\pi^2}\delta^{ab}\partial^\mu\delta(x-y)j_{\mu\nu}^{(2)}(y)\rangle\,, \quad n \in \mathbb{Z}\,, \quad (1.1)$$

where $f^{abc}$ are the structure constants for the 0-form flavour symmetry. The fact that the coefficient $n$ is an integer is reminiscent of the original anomalous current algebra relation. (Here $n \in \mathbb{Z}$ really labels the 'Postnikov class' characterizing the particular 2-group, and is an element in an appropriate cohomology group.) Indeed, like the anomaly, it implies that such a 2-group structure should be preserved along the RG flow, or else the symmetry be broken.

This paper concerns a seemingly innocuous extension of 4d QCD (let's say, for concreteness, that the UV theory contains an $\mathrm{SU}(n_c)$ gauge group and $N$ fundamental

Dirac fermions) by a pair of extra pions in the infrared that live on $S^2$. This theory admits a second topological term in the low-energy effective action that involves both QCD pions and the extra pions on $S^2$, as recently observed in Ref. [14]. Like the original WZW term, this topological term has an integer-quantized coefficient. But unlike the WZW term, this term does not match any chiral anomaly – so it has no obvious interpretation as originating from a loop of chiral fermions in the microscopic theory.

In this paper, we show that this 'mixed WZW term' actually encodes a 2-group global symmetry structure in the IR theory, which mixes the QCD flavour symmetry with a 1-form symmetry associated to the winding number around the $S^2$ factor of the target space. The coefficient $n$ of this mixed WZW term is precisely the Postnikov class encoding the 2-group symmetry. This furnishes an interesting new example of continuous 2-group symmetry in a 4d quantum field theory, that contains only scalar degrees of freedom in the infrared.[1]

The observation of a non-trivial 2-group global symmetry in this theory puts strong constraints on possible RG flows that can realise this phase in the IR.[2] In particular, a UV completion that exactly preserves the QCD flavour symmetry must, in order to close the 2-group current algebra relation (1.1), at the very least possess a continuous 1-form symmetry. At a stroke, this precludes QCD-like UV completions that one might guess, such as a confining SU × SO gauge theory or an SU gauge theory with both fundamental and adjoint quarks condensing.

We then propose a UV completion of this mixed WZW term, consistent with the 2-group symmetry constraints, in which an abelian $U(1)_g$ gauge field that is Higgsed in the IR connects the QCD sector to the $S^2$ sector. The scalars on $S^2$ are embedded in a linear sigma model of two complex scalars at high energy, charged equally under $U(1)_g$, which condense to Higgs the $U(1)_g$ and simultaneously break a global SU(2) symmetry down to a U(1) subgroup, delivering the massless pions on $\mathrm{SU}(2)/\mathrm{U}(1) \cong S^2$. Once the abelian gauge field is 'integrated in' as we go to higher energies, the $S^2$ winding number is precisely traded for the abelian monopole flux. On the quark side, the $U(1)_g$ gauge field couples to baryon number, which becomes identified with the topologically conserved Skyrme current [18–20] in the IR. These very particular couplings of the $U(1)_g$ gauge field are the ingredients needed to match onto the mixed WZW topological term, simply by integrating out a weakly coupled gauge field at tree-level. Notably, because the coefficient of the mixed WZW term is quantized for consistency of the low-energy effective action (as mentioned, it is a class in integral cohomology), this tree-level matching formula cannot receive loop correction. In other words, the matching must be tree-level exact.

---

[1]Other examples of continuous 2-group symmetry without fermions include continuous 2-groups in hydrodynamics and holography [15, 16].

[2]The rigidity of higher-group symmetry structures mean they can also be tracked across dualities, as well as RG flows, to provide highly non-trivial checks – see for instance [17].

When an anomaly-free abelian subgroup of the QCD flavour symmetry is also gauged (such as gauging QED in real-world QCD), the symmetry structure becomes more intricate. We show how additional non-invertible symmetries arise (beyond those corresponding to the 'usual' ABJ anomaly in pure QCD), while a remnant of the original 2-group global symmetry remains.

Finally, we remark that the identification of this peculiar mixed WZW term in Ref. [14] was motivated by the fact that it can play an important role in phenomenology. It was there shown that the extra pions on $S^2$ could constitute the dark matter (DM) in our Universe, with the mixed WZW term providing an (almost) unique portal from this dark sector to low-energy QCD that is topological.[3] This portal can reproduce the observed abundance of dark matter today via thermal freeze-out.[4] The present paper offers one path to UV completing this novel portal, guided by the identification of a rich generalised symmetry structure, that will be crucial to understanding the full phenomenological implications (both in cosmology and collider experiments) of this topological portal EFT.

The rest of the paper is as follows. In §2 we recall the construction of the mixed WZW term in the low-energy EFT of QCD extended by pions on $S^2$, and we identify the presence of 2-group symmetry encoded by this term. We then use this symmetry structure to investigate the UV completion of this EFT: in §3 we discuss QCD-like 'non-completions', before setting out the weakly coupled UV completion in §4 wherein QCD is coupled to a linear sigma model by an abelian gauge field. We consider variations of the scalar sector, in particular a generalisation of our story obtained by replacing the $S^2$ factor by a general complex projective space, in §5. Finally, in §6 we treat the gauged case, before concluding.

## 2 Mixed WZW term on $\mathrm{SU}(N) \times S^2$

Our story starts with a low-energy effective field theory (EFT) of pions on the manifold $\mathrm{SU}(N) \times S^2$, in 3+1 dimensions. This would arise from an ultraviolet theory with approximate global symmetry of the product form $G = \mathrm{SU}(N)_L \times \mathrm{SU}(N)_R \times \mathrm{SU}(2)_D$, which is spontaneously broken down to $H = \mathrm{SU}(N)_{L+R} \times \mathrm{U}(1)_D$ as we flow to

---

[3]Other WZW-like topological portals between low-energy QCD + electromagnetism and a dark sector could (a) connect QCD baryon number to a dark photon, or (b) connect the visible photon to a dark baryon number current. There could be yet further topological portals that are, however, more like $\theta$-terms, for instance mixing the visible photon with a dark photon; but these would be total derivatives and so not give rise to local interactions between the visible and dark particles.

[4]The topological nature of the interaction is not just a theoretical nicety, but plays a crucial role in the phenomenology: being anti-symmetrized in field indices completely suppresses elastic interaction channels that would otherwise lead to strong constraints from DM direct and indirect detection. This allows the off-diagonal 'co-annihilation' channel to dominate, realising the light thermal inelastic DM scenario [21].

the infrared, where $\mathrm{SU}(N)_{L+R}$ is the diagonal subgroup of $\mathrm{SU}(N)_L \times \mathrm{SU}(N)_R$ and $\mathrm{U}(1)_D \subset \mathrm{SU}(2)_D$.

(Co)homological [22] (or (co)bordism-based [23, 24]) classifications of Wess–Zumino–Witten (WZW) terms [5, 9] tell us there is a *mixed* WZW interaction coming from the existence of a $G$-invariant closed 5-form involving pions on both the $\mathrm{SU}(N)$ and $S^2$ factors, namely [14]

$$\omega = n\omega_3 \wedge \mathrm{Vol}_{S^2}, \qquad \omega_3 = \frac{1}{24\pi^2}\mathrm{Tr}(g^{-1}dg)^3\,, \qquad n \in \mathbb{Z}\,, \tag{2.1}$$

where $\mathrm{Vol}_{S^2}$ is the volume form on the $S^2$. To define this term it is easier to start with a homological description (we briefly discuss the refinement via bordism afterwards). Given there are no homologically non-trivial 4-cycles in the target space, since $H_4(\mathrm{SU}(N) \times S^2) = 0$, any 4-cycle in the target space obtained by pushing forward $\Sigma_4$ (more precisely, pushing forward a cycle in the fundamental class $[\Sigma_4]$) can be realised as the boundary of a 5-cochain $X_5$, on which we can integrate $\omega$ to obtain the exponentiated action à la Witten [9]

$$\exp\left(iS[\Sigma_4 = \partial X_5]\right) = \exp\left(2\pi i \int_{X_5} \omega\right) \tag{2.2}$$

$$= \exp\left(2\pi i \int_{X_5} \frac{n}{24\pi^2}\mathrm{Tr}(g^{-1}dg)^3 \wedge \mathrm{Vol}_{S^2}\right)\,.$$

The normalisation is such that $\omega$ is an integral 5-form on $\mathrm{SU}(N) \times S^2$, meaning that $e^{2\pi i \int_{z_5} \omega} = 1$ for any 5-cycle $z_5$, which guarantees the exponentiated action defined in this way is independent of the choice of 'bulk manifold' (more precisely, 5-cochain) $X_5$ [9].

A local expression for the 4d Lagrangian can be obtained by expanding the QCD pion field locally as

$$g(x) = \exp(2i\pi_a(x)t^a/f_\pi) = 1 + 2i\pi_a(x)t^a/f_\pi + \dots\,, \tag{2.3}$$

and taking $\chi_i/f_D$ as local Cartesian coordinates in the vicinity of a given vacuum point ($\chi_i = 0$) on the $S^2$ factor. Then the $\mathrm{SU}(N)$-invariant 3-form is expanded as

$$\omega_3 = \frac{1}{24\pi^2}\frac{2}{f_\pi^3}f_{abc}d\pi_a \wedge d\pi_b \wedge d\pi_c + \mathcal{O}(\pi^4)\,, \tag{2.4}$$

where $f_{abc}$ are the $\mathrm{SU}(N)$ structure constants, while the volume form on the $S^2$ is expressed as

$$\mathrm{Vol}_{S^2} = \frac{1}{4\pi}\frac{1}{f_D^2}\cos(\chi_1)d\chi_1 \wedge d\chi_2 = \frac{1}{4\pi}\frac{1}{f_D^2}d\chi_1 \wedge d\chi_2 + \mathcal{O}(\chi^3)\,. \tag{2.5}$$

On the coordinate patch near the origin $(\pi_a, \chi_i) = 0$, we can use Stokes' theorem to get a local expression for the Lagrangian (also including a factor $2\pi i$):

$$L = \frac{in\epsilon^{\mu\nu\rho\sigma}}{48\pi^2 f_D^2 f_\pi^3}f_{abc}\epsilon_{ij}\pi_a\partial_\mu\pi_b\partial_\nu\pi_c\partial_\rho\chi_i\partial_\sigma\chi_j + \mathcal{O}(\pi^4\chi^2, \pi^3\chi^3)\,, \tag{2.6}$$

which was used to calculate scattering cross-sections between the $\pi$ and $\chi$ pions in [14].

## 2.1 Remarks on bordism *vs* homology

To be more precise, one should re-formulate the above construction of a WZW-like action, which involves realising spacetime $\Sigma$ as a boundary of a bulk in one higher dimension, more directly using the language of bordism. This is somewhat tangential to the main story of this paper, so we limit ourselves to some cursory remarks.

Roughly speaking, bordism tells us whether manifolds equipped with certain structures (gauge bundles, spin structures, metrics) can be written as boundaries of manifolds in one dimension higher, with all structures smoothly extended thereto. In short, bordism deals directly with manifolds themselves (rather than passing to cochains as in the homological description above). The UV theory we have in mind will feature fermions, so the appropriate bordism theory is (reduced) spin-bordism.

In Appendix A we use the Adams spectral sequence [25] to compute that the relevant reduced bordism group for us here does not vanish, but is given by

$$\tilde{\Omega}_4^{\mathrm{Spin}}\left(\mathrm{SU}(N) \times S^2\right) \cong \mathbb{Z}_2\,. \tag{2.7}$$

This means there is a class of spin 4-manifolds $[X_4]$, together with maps $\sigma$ to $\mathrm{SU}(N) \times S^2$, that cannot be extended to spin 5-manifolds (also equipped with maps to $\mathrm{SU}(N) \times S^2$); this equivalence class is the generator of the bordism group (2.7). The existence of manifolds that are not boundaries is an obstruction to a Witten-like definition (2.2) of the WZW term when evaluated on such manifolds. This obstruction is not seen using homology which, amongst other differences, is not sensitive to spin structures and whether or not they can be extended also.

We can nonetheless proceed in attempting to define our WZW term on a general manifold as follows. That the bordism group is $\mathbb{Z}_2$ means that the union of two generators is a boundary of some 5-manifold $X_5$ with the map $\sigma$ extended. The exponentiated action evaluated on this union $X_4 \sqcup X_4$ must be the square of the action evaluated on the original $X_4$, by locality. Therefore, we can define the exponentiated WZW action for this union as

$$(\exp(iS[X_4]))^2 = \exp\left(2\pi i \int_{X_5} \omega\right) \tag{2.8}$$

There are then two possibilities for the WZW term for the generator of the bordism group, differing by a sign. This sign can be fixed by anomaly matching: the solution with a minus sign must be chosen if the symmetry group $\mathrm{SU}(2)$ suffers from the mod 2 global anomaly discovered by Witten [26] in the UV. For more detail on matching global anomalies with WZW terms, see Refs. [23, 24].

In principle there is also a subtlety concerning the normalisation of the WZW term, related to whether one uses a classification based on homology, bordism, or

even homotopy groups.[5] In our case, the normalisation encoded in our formulae (2.1) above is that prescribed by homology; nonetheless, later we show that this mixed WZW term relates to an anomaly polynomial (4.9) for a particular fermionic theory, whose normalisation condition is determined by the Atiyah–Singer index theorem [29–31]. This verifies that the normalisation of (2.1) coincides with the correct normalisation in spin bordism.

## 2.2 An anomaly-matching puzzle

We now continue with the main line of argument, and consider the symmetry/anomaly information encoded in this mixed WZW term. Recall the coefficient of the ordinary WZW term for pure QCD, associated with the 5-form $\omega_5 \propto \text{Tr}(g^{-1}dg)^5$, is fixed by 't Hooft anomaly matching [32] for the anomalous chiral symmetry currents generating $\text{SU}(N)_L$ and $\text{SU}(N)_R$ separately. The UV anomalies come from terms in the anomaly polynomial $\Phi_6 \sim \text{Tr}(F_L^3) + \text{Tr}(F_R^3)$. When QED is gauged, this translates into an ABJ anomaly that explicitly breaks the chiral flavour symmetries.

We can similarly investigate the possible anomaly-matching role of the mixed WZW term by turning on gauge currents for various global symmetries. For instance, upon gauging QED one also gets the term

$$L \sim \frac{1}{f_\pi f_D^2} \epsilon^{\mu\nu\rho\sigma} \pi_0 F_{\mu\nu} \partial_\rho \chi_1 \partial_\sigma \chi_2 \,, \tag{2.9}$$

which was put to phenomenological use as a possible portal to dark matter in [14]. If one also gauged the dark $\text{U}(1)_D$ unbroken global symmetry, one would further get a term

$$L \sim \frac{1}{f_\pi} \epsilon^{\mu\nu\rho\sigma} \pi_0 F_{\mu\nu} (F_D)_{\rho\sigma} \,. \tag{2.10}$$

Thus, the mixed WZW term naïvely matches mixed anomalies corresponding to anomaly polynomial terms $\Phi_6 \sim a_L \text{Tr}(F_L^2 F_D) + a_R \text{Tr}(F_R^2 F_D)$. One might try to conclude that this EFT term arises in UV theories that feature a mixed anomaly between the 'light' and 'dark' flavour symmetries. But these mixed anomaly coefficients $a_{L/R}$ *vanish* if the $\text{U}(1)_D$ representations are embedded in $\text{SU}(2)_D$, because there is of course no mixed anomaly between $\text{SU}(2)$ and $\text{SU}(N \geq 3)$ in 4d. The mixed WZW term in the low-energy EFT is consistent with an exact $\text{SU}(2)_D$ flavour symmetry, and so there should be a UV account of its origin that does not invoke explicit $\text{SU}(2)_D$-breaking as a necessary ingredient.

The main question we try to answer in this paper is: what is the microscopic origin, if not an anomaly, of the mixed WZW term permitted by this particular EFT?

---

[5]For example, the normalisation of the usual WZW term of 4d QCD is subject to a factor of two difference [27] if one chooses to normalise the curvature form $\propto \text{Tr}\,(g^{-1}dg)^5$ against the generator of the homotopy group $\pi_5(\text{SU}(N)) = \mathbb{Z}$ or against the integral homology $H_5(\text{SU}(N); \mathbb{Z}) = \mathbb{Z}$; furthermore, the homotopy-based normalisation happens to agree with that determined by spin-bordism [23, 28], which is arguably the most justifiable choice.

## 2.3 Infrared 2-group symmetry

Generalised symmetries (in place of anomalies) present a key to unlocking this apparent puzzle. It turns out that the mixed WZW signals the presence of a generalised symmetry structure known as a *2-group symmetry*, which mixes the QCD flavour 0-form symmetry with a 1-form global symmetry in a non-trivial way.

First, we observe that the volume form on $S^2$ provides us with a topologically conserved (*i.e.* closed) non-trivial 2-form, which may be identified with the current for a 1-form winding number symmetry

$$j_{\text{wind}}^{(2)} := \star_4 \text{Vol}_{S^2} = \frac{\sqrt{|\det(g)|}}{4\pi} \epsilon_{\mu\nu\rho\sigma} \cos(\chi_1) \partial^\mu \chi_1 \partial^\nu \chi_2 dx^\rho dx^\sigma, \tag{2.11}$$
$$d \star_4 j_{\text{wind}}^{(2)} = d \,\text{Vol}_{S^2} = 0 \,.$$

Here, $\star_4$ is the Hodge dual on spacetime $(\Sigma, g)$, and indices have been raised using the inverse metric on $\Sigma$, *viz.* $\partial^\mu \chi_i = g^{\mu\lambda} \partial_\lambda \chi_i$.

To understand the global symmetry structure precisely, it is instructive to turn on background gauge fields. Let us start with the 1-form symmetry. As for a regular 0-form symmetry, one can minimally couple the 1-form symmetry to a background gauge field, which in this case is a 2-form gauge field $B$. The minimal coupling term is

$$S_{\text{coup}} = i \int_{\Sigma_4} \star_4 j_{\text{wind}}^{(2)} \wedge B = i \int_{\Sigma_4} \text{Vol}_{S^2} \wedge B \tag{2.12}$$

By construction, the coupling term is invariant under a $\text{U}(1)_{\text{wind}}^{[1]}$ 1-form gauge transformation

$$B \mapsto B + d\Lambda^{(1)}, \tag{2.13}$$

because $\text{Vol}_{S^2}$ is closed.

Now for the 0-form symmetry, we turn on background 1-form gauge fields $A_L$ and $A_R$. The gauging of these 0-form symmetries in the mixed WZW term can be deduced from the gauging of the 3d WZW term $S[\Sigma_2] := \int_{Y_3} \text{Tr}\,(g^{-1}dg)^3/24\pi^2$, where $\partial Y_3 = \Sigma_2$ which we know yields the 3d Chern–Simons (CS) theory $\int_{Y_3} \frac{1}{8\pi^2} [\text{CS}(A_L) - \text{CS}(A_R)]$ where $\text{CS}(A) = AdA + \frac{2}{3}A^3$. For our 4d theory, gauging therefore gives the action

$$iS[\Sigma_4 = \partial X_5] = \int_{X_5} \frac{-i}{8\pi^2} [\text{CS}(A_L) - \text{CS}(A_R)] \wedge \text{Vol}_{S^2} \tag{2.14}$$

Crucially, after coupling to the background gauge fields $A_L$, $A_R$, $B$ for $\text{SU}(N)_L$, $\text{SU}(N)_R$, and $\text{U}(1)_{\text{wind}}^{[1]}$, respectively, the mixed WZW term is not gauge invariant. Rather, under the flavour 0-form gauge transformation

$$A_L \mapsto A_L + D_{A_L}\lambda_L^{(0)}, \qquad A_R \mapsto A_R + D_{A_R}\lambda_R^{(0)}, \tag{2.15}$$

it shifts by

$$\delta(2\pi i S) = -\frac{in}{4\pi} \int_{\Sigma_4} \text{Vol}_{S^2} \wedge \left[ \text{Tr}\,(\lambda_L^{(0)} dA_L) - \text{Tr}\,(\lambda_R^{(0)} dA_R) \right]. \tag{2.16}$$

One might interpret this as an extra anomalous variation under QCD flavour symmetry, in addition to the anomalous variation already encoded in the usual WZW term of pure QCD. But an important difference compared to the usual WZW term variation is that, for this term, the factor $\text{Vol}_{S^2}$ on the RHS of the anomalous variation is an operator in the theory that does not vanish upon turning off the 0-form background gauge field.

However, now consider modifying the transformation law for the 2-form gauge field to depend on the QCD flavour 0-form transformation, in the following way:

$$B \mapsto B + d\Lambda^{(1)} + \frac{\hat{\kappa}_L}{4\pi} \text{Tr} \left( \lambda_L^{(0)} dA_L \right) + \frac{\hat{\kappa}_R}{4\pi} \text{Tr} \left( \lambda_R^{(0)} dA_R \right) . \tag{2.17}$$

Mathematically, this (together with 2.15) defines the infinitesimal version of a continuous 2-group gauge transformation,[6] with the locally-defined differential forms $(A_L, A_R, B)$ forming a 2-connection. Given the 0-form flavour symmetry $\text{SU}(N)^2$ and the 1-form winding number symmetry valued in $\text{U}(1)^{[1]}$, possible 2-group structures intertwining the two are classified (subject to certain simplifying assumptions that apply here) by a topological invariant called the *Postnikov class*, which is the pair

$$(\hat{\kappa}_L, \hat{\kappa}_R) \in H^3(B\text{SU}(N)^2; \text{U}(1)) \cong \mathbb{Z} \times \mathbb{Z} . \tag{2.18}$$

If we postulate this generalised gauge transformation law for the background fields, then the minimal coupling term (2.12) shifts by

$$\frac{i}{4\pi} \int_{\Sigma_4} \text{Vol}_{S^2} \wedge \left[ \hat{\kappa}_L \text{Tr} \left( \lambda_L^{(0)} dA_L \right) + \hat{\kappa}_R \text{Tr} \left( \lambda_R^{(0)} dA_R \right) \right] . \tag{2.19}$$

This is precisely how the mixed WZW term with the $SU(N)_{L/R}$ background gauge fields turned on shifts, in the opposition direction, under the flavour gauge transformation, provided that we identify

$$\hat{\kappa}_L = -\hat{\kappa}_R = n . \tag{2.20}$$

The whole combination is then gauge invariant. This is a sign that there is a nontrivial 2-group structure present [34], and that the coefficient of the mixed WZW term exactly determines the Postnikov class characterizing the 'twisting' of this 2-group symmetry.

### Ward identities for the 2-group symmetry

To see that this is not just an artifact of turning on background fields, we can show how this 2-group structure is manifest at the level of current algebra. We start by

---

[6]We will not actually define what a 2-group is in this paper. For an introduction to the mathematical notion of a 2-group, and how these structures appear in QFT, see *e.g.* §2 of [33].

deriving the modified conservation laws for the 1-form currents when only the background fields for the flavour symmetries are turned on. Recall that the currents $j_L^{(1)}$ and $j_R^{(1)}$ for the flavour symmetries $\mathrm{SU}(N)_L$ and $\mathrm{SU}(N)_R$ can be minimally coupled to their corresponding background gauge fields as

$$S_{\text{currents}} = \int_{\Sigma_4} \left( \mathrm{Tr} \, \star j_L^{(1)} \wedge A_L + \mathrm{Tr} \, \star j_R^{(1)} \wedge A_R \right). \tag{2.21}$$

Including also the mixed WZW term, and using the gauge transformation of the mixed WZW term given in Eq. (2.16), the total action shifts by

$$\delta S = \int_{\Sigma_4} \left[ \left( D_{A_L} \star j_L^{(1)} - \frac{n}{8\pi^2} \star j_{\text{wind}}^{(2)} \wedge dA_L \right)^a \lambda_L^a \right. \tag{2.22}$$
$$\left. + \left( D_{A_R} \star j_R^{(1)} + \frac{n}{8\pi^2} \star j_{\text{wind}}^{(2)} \wedge dA_R \right)^a \lambda_R^a \right],$$

upon integrating by parts the terms involving 1-form currents. Imposing gauge invariance, we obtain a pair of modified conservation laws:

$$D_{A_L} \star j_L^{(1)} = \frac{n}{8\pi^2} \star j_{\text{wind}}^{(2)} \wedge dA_L, \tag{2.23}$$

$$D_{A_R} \star j_R^{(1)} = -\frac{n}{8\pi^2} \star j_{\text{wind}}^{(2)} \wedge dA_R, \tag{2.24}$$

which hold inside the path integral. This means

$$\int \mathcal{D}\Phi \left( D_{A_L} \star j_L^{(1)} - \frac{n}{8\pi^2} \star j_{\text{wind}}^{(2)} \wedge dA_L \right) e^{iS_{\text{current}}} e^{iS_0} = 0, \tag{2.25}$$

where $S_0$ is the original action when all background fields are turned off, and $\Phi$ stands for all the dynamical fields collectively. A similar expression holds for $j_R^{(1)}$.

Current algebra relations for $j_L^{(1)}$ and $j_R^{(1)}$ can then be obtained by taking functional derivatives of these conservation laws with respect to the background fields. Expanding the integrand of the path integral above for infinitesimal $A_L$ (with $A_R = 0$), we obtain, at linear order,

$$i\partial_\mu j_L^{(1)a\mu}(x) \int d^4y A_{L\nu}^b(y) j_L^{(1)b\nu}(y) + f^{abc} A_L^b(x) j_L^{c(1)\nu}(x) = \frac{n}{8\pi^2} j_{\text{wind}}^{(2)\lambda\nu}(x) \partial_\lambda A_{L\nu}^a(x) \tag{2.26}$$

Taking the functional derivative of this equation with respect to $A_{L\nu}^b(y)$ gives

$$i\partial_\mu j_L^{(1)a\mu}(x) j_L^{(1)b\nu}(y) + f^{abc} \delta(x-y) j_L^{(1)c\nu}(y) = \frac{n}{8\pi^2} \delta^{ab} \frac{\partial}{\partial x^\lambda} \delta(x-y) j_{\text{wind}}^{(2)\lambda\nu}(y). \tag{2.27}$$

Repeating the operation with $A_R$, we similarly obtain

$$i\partial_\mu j_R^{(1)a\mu}(x) j_R^{(1)b\nu}(y) + f^{abc} \delta(x-y) j_R^{(1)c\nu}(y) = -\frac{n}{8\pi^2} \delta^{ab} \frac{\partial}{\partial x^\lambda} \delta(x-y) j_{\text{wind}}^{(2)\lambda\nu}(y). \tag{2.28}$$

Sure enough, these expressions exactly reproduce the current algebras for the non-abelian 2-group structure between $SU(N)_L \times SU(N)_R$ 0-form symmetry and $U(1)_{\text{wind}}$ 1-form symmetry, with Postnikov classes $\hat{\kappa}_L = n$ and $\hat{\kappa}_R = -n$ [34].

To summarise so far, we have shown that the IR mixed WZW term implies a non-trivial 2-group structure, with $U(1)$ as the 1-form part. In the following Sections, we will use this 2-group structure as a guide to build a UV completion for this EFT.

# 3  From infrared to ultraviolet: a no-go theorem

One might try to UV complete this 'product coset' model with a QCD-like strongly coupled gauge theory, with two types of quark field that confine at different scales to give the two types of pions in the IR. We have already seen that such a UV completion does not fix the mixed WZW term coefficient by anomaly matching, because there is no non-abelian mixed anomaly in the UV. Now we will show something even stronger, which is that the mixed WZW term in the IR is in fact *inconsistent* with the QCD-like dark sector completion! We will then build a UV completion that does work in §4.

To see why a QCD-like dark sector cannot generate this mixed WZW term, let's have a concrete model in mind. To get the symmetry breaking pattern $SU(2)_D \to U(1)_D$, a candidate dark dynamics is $SO(N_c)$ gauge theory with two flavours of fundamental dark quark. Including also the QCD part, which takes the form of an $SU(n_c)$ gauge theory acting on fundamental quarks, and allowing for other matter fields transforming in linear representations under both $SO(N_c)$ and $SU(n_c)$ that communicate weakly between the two sectors, there is no continuous $U(1)^{[1]}$ 1-form symmetry in this phase (although there is a $\mathbb{Z}_2^{[1]}$ 1-form symmetry associated with the gauged SO group).

But the absence of a 1-form symmetry means the current algebra for the 0-form QCD symmetries does not close; recall the Postnikov class appearing in the 2-group current algebra relations (2.27, 2.28) is integer-quantized, and so preserved under RG flow. Put in a more general context, the problem is that the IR theory (with non-zero mixed WZW term) encodes a non-trivial *extension* between the flavour symmetry and the 1-form symmetry, manifest in the fibration

$$BU(1) \hookrightarrow \mathbb{G}_{\text{IR}} \to SU(N)_L \times SU(N)_R \qquad (3.1)$$

being topologically non-trivial (where here $\mathbb{G}_{\text{IR}}$ denotes the infrared 2-group symmetry we have detected). This means that, if only the 0-form $SU(N)^2$ flavour symmetry is there in the UV, then the 1-form symmetry cannot be emergent and end up twisted in such a non-trivial fibration.[7] As a result, this kind of dynamics *cannot* possibly

---

[7]The same argument 'against infrared emergence' should apply in other contexts, not just for 2-group symmetry but for any infrared symmetry that is a non-trivial extension. We thank Y. Tachikawa for raising this point.

UV complete our IR EFT with mixed WZW term, unless the UV also breaks the 0-form flavour symmetry.

We thus establish a 'no-go theorem':

> In the absence of any explicit symmetry breaking, the sigma model on $G/H \times \mathrm{SU}(2)/\mathrm{U}(1)$ with non-zero mixed WZW term cannot be UV completed by a non-abelian gauge theory with semisimple gauge group.

A valid UV completion that preserves the 0-form QCD flavour symmetries *must*, at the very least, possess a non-trivial 1-form symmetry that can close the 2-group operator algebra.

One can refine this picture to consider UV theories in which both the 0-form flavour symmetry and the winding number 1-form symmetry are emergent in the infrared. The *2-group emergence theorem*[8] of Ref. [34] then implies the scale hierarchy $\Lambda_{\text{flavour}} \lesssim \Lambda_{\text{1-form}}$, where $\Lambda_{\text{flavour}}$ ($\Lambda_{\text{1-form}}$) denotes the scale where the 0-form QCD flavour symmetry (1-form symmetry) emerges. Assuming the 0-form symmetry remains an exact symmetry of the UV (as in a QCD-like completion with explicit symmetry breaking sources turned off) corresponds to the limit $\Lambda_{\text{flavour}} \to \infty$.

## 4 From infrared to ultraviolet: a weakly-coupled completion

Informed by this observation, we propose that the mixed WZW term arises from a particular coupling between QCD and scalar electrodynamics that enables a non-trivial 2-group structure. The 0-form part of the 2-group structure is the $\mathrm{SU}(N)_L$ (and, separately, $\mathrm{SU}(N)_R$) flavour symmetry, while the 1-form part is the $\mathrm{U}(1)$ magnetic symmetry associated with the $\mathrm{U}(1)$ gauge symmetry of the scalar electrodynamics.

We stress that it is crucial that the extra gauge symmetry be abelian to bestow us with a candidate 1-form symmetry, with which we can try to close the 2-group current algebra in the UV and thence match the IR symmetries encoded in the mixed WZW term.

### 4.1 The UV phase: QCD coupled to scalar electrodynamics

The particular scalar electrodynamics (SED) that we will couple to the QCD sector consists of two complex scalar fields $\phi_1$ and $\phi_2$, both coupled to the $\mathrm{U}(1)$ gauge field $b$ with charge $+1$. The dynamics of our SED is governed by the Lagrangian

$$\mathcal{L}_{\text{SED}} = -\frac{1}{4e^2}(db)^2 + \sum_{i=1}^{2}|(\partial - ib)\phi_i|^2 + m^2\sum_{i=1}^{2}|\phi_i|^2 + \lambda\sum_{i=1}^{2}|\phi_i|^4, \qquad (4.1)$$

---

[8]This 2-group emergence theorem has been applied, for instance, to study models for unification in [35] and, perhaps more closely to the present work, to theories with axions in [36, 37].

where $e$ is the gauge coupling, $m^2$ is the mass squared parameter for the scalars (of as-yet unfixed sign), and $\lambda$ is the parameter for the quartic potential. Without the gauging, the scalar potential would have an O(4) accidental symmetry, exactly like the Higgs sector of the electroweak theory with gauge fields turned off. When the U(1) is gauged, akin to gauging hypercharge in the electroweak theory, the remaining accidental global symmetry is reduced to an SU(2) $\subset$ O(4).[9]

We then couple the quarks to the SED sector through the U(1) gauge field, which we also take to have charge $+1$ for now.[10] The full Lagrangian is then given by

$$\mathcal{L}_{\text{full}} = \mathcal{L}_{\text{QCD}} + \mathcal{L}_{\text{SED}} + \mathcal{L}_{\text{int}}, \tag{4.2}$$

where

$$\mathcal{L}_{\text{QCD}} = -\frac{1}{2g^2}\text{Tr}\ f_{\mu\nu}f^{\mu\nu} + \sum_{i=1}^{N} i\overline{\Psi}_i \left(\slashed{\partial} - i\slashed{a}\right)\Psi_i, \tag{4.3}$$

$$\mathcal{L}_{\text{int}} = \sum_{i=1}^{N} \overline{\Psi}_i \gamma^\mu b_\mu \Psi_i \tag{4.4}$$

Here, $a$ denotes the SU($n_c$) colour gauge field, and $f = da - ia \wedge a$ its field strength.

## 4.2 Symmetries and anomalies of the UV theory

The faithfully acting global symmetry of this theory appears to be

$$G_{\text{glob}} = \frac{\text{U}(1)_q \times \text{SU}(N)_L \times \text{SU}(N)_R \times \text{SU}(2)_\phi}{\mathbb{Z}_{n_c} \times \mathbb{Z}_N \times \mathbb{Z}_2} \times \text{U}(1)_m^{[1]}, \tag{4.5}$$

To see this, let us first enumerate 0-form symmetries. As in the usual massless QCD, there are flavour 0-form symmetries SU($N$)$_L$ and SU($N$)$_R$ that acts on the left-handed and right-handed components of $\Psi$ independently. As already remarked, there is an SU(2) flavour symmetry for the scalars acting by

$$\text{SU}(2)_\phi: \qquad \phi_i \mapsto U_i{}^j \phi_j, \quad U \in \text{SU}(2). \tag{4.6}$$

Lastly, despite coupling the quarks to a U(1) gauge symmetry, the U(1)$_q$ quark symmetry

$$\text{U}(1)_q: \quad \Psi_i \mapsto e^{i\alpha}\Psi_i, \tag{4.7}$$

remains a good global symmetry. This is because the U(1) gauge symmetry acts on both the quarks and the scalars, leaving the rotations on the quarks alone independent. This matches our accounting for the gauged U(1) in the scalar part above.

---

[9]In the case of the electroweak theory, but not here, this SU(2) is of course also gauged.

[10]We shall consider the generalisation in which the quarks and scalars have different charge, which actually introduces topological complications, in §5.

The quotient by $\mathbb{Z}_{n_c}$ in Eq. (4.5) reflects the fact that the subgroup $\mathbb{Z}_{n_c} \subset$ U(1)$_q$ can be rotated away by a SU($n_c$) gauge transformation. On the other hand, the $\mathbb{Z}_N$ quotient is there to avoid double counting, because the action of the $\mathbb{Z}_N$ subgroup of the diagonal SU($N$) $\subset$ SU($N$)$_L \times$ SU($N$)$_R$ is the same as the action of the $\mathbb{Z}_N$ subgroup of U(1)$_q$. The quotient by $\mathbb{Z}_2$ is due to the fact that the centre of U(1)$_q \times$ SU(2)$_\phi$ coincides with a $\mathbb{Z}_2$ subgroup of the U(1) gauge group, and thus can be gauged away.

In addition to these 0-form symmetries, there is a magnetic 1-form symmetry U(1)$_m^{[1]}$ acting on 't Hooft line defects, whose conserved charges measure the magnetic fluxes of these defects. The symmetry inevitably arises in a U(1) gauge theory. Because the U(1) field strength $h := db$ is closed, $dh = 0$, it naturally defines a 2-form current $j_m^{(2)} := \star h/2\pi$ which is conserved,

$$d \star j_m^{(2)} = \frac{dh}{2\pi} = 0 \,, \tag{4.8}$$

even before imposing the equations of motion. It is therefore an example of a topologically-conserved 1-form symmetry. Notice that this enjoys the same status as the topologically-conserved 2-form $j_{\rm wind}^{(2)}$ that we previously identified in the IR sigma model.

This is not the whole story. The magnetic 1-form symmetry and the flavour symmetries form a non-trivial 2-group structure. The Postnikov class which characterises the structure can be read-off directly from the associated anomaly polynomial of the theory once we turn on the background gauge fields $A_{L/R}$ for SU($N$)$_{L/R}$, with corresponding field strength 2-forms $F_{L/R}$. The degree-6 anomaly polynomial is given by

$$\Phi_6 = \frac{n_c}{3!} \frac{1}{(2\pi)^3} \left[ {\rm Tr}\, F_L^3 - {\rm Tr}\, F_R^3 \right] + \frac{n_c}{2} \frac{h}{2\pi} \left[ {\rm Tr}\, \left(\frac{F_L}{2\pi}\right)^2 - {\rm Tr}\, \left(\frac{F_R}{2\pi}\right)^2 \right] \tag{4.9}$$

The first term represents the usual 't Hooft anomaly for the SU($N$)$_{L/R}$ chiral global symmetries, that is matched in the IR by the familiar WZW term of pure QCD constructed from ${\rm Tr}\, (g^{-1}dg)^5$.

The second term represents the 'operator-valued mixed anomalies' between SU($N$)$_{L/R}$ and the U(1) gauge symmetry, and can be properly interpreted as intertwining the 0-form symmetries SU($N$)$_{L/R}$ with the 1-form magnetic symmetry to form a non-trivial 2-group structure [34]. The Postnikov classes characterising these 2-group structures are given by the anomaly coefficient as

$$\hat{\kappa}_L = -\hat{\kappa}_R = n_c \,. \tag{4.10}$$

Comparing with the 2-group structure enshrined by the mixed WZW term in the IR, which recall was captured by Postnikov classes in Eq. (2.20), suggests that matching

of the global symmetries fixes the coefficient of the mixed WZW term in the IR:

$$n = n_c \,, \qquad (4.11)$$

in very close analogy to the coefficient of the ordinary WZW term that is fixed by the 't Hooft anomaly!

Of course, we have not yet shown that this theory actually UV completes the IR sigma model with mixed WZW term; all we have so far shown is that the generalised global symmetries we identified (and the 't Hooft anomalies, as inherited from the pure QCD part) all match. In the next few Sections, we explicitly follow the RG flow, starting from this UV theory, to show how we arrive at the mixed WZW terms after going through two phase transitions.

## 4.3 The Higgs phase

To begin, we first define the parameters in the scalar sector (with a choice of sign for the quadratic term) so that the potential can be written

$$V(\phi_i) = \lambda \left( |\phi_i|^2 - v^2 \right)^2 \,, \qquad (4.12)$$

so that the scalars acquire a non-zero vacuum expectation value (VEV). Without turning on the U(1) gauge field $b$, the vacuum manifold would be $S^3$, given by the minima of the potential. This reflects the symmetry breaking pattern O(4) → O(3). Taking the U(1) quotient from the gauge group reduces the vacuum manifold down to $S^2$.

To simplify matters it is helpful to stagger the phase transitions in the QCD sector and the scalar sector, so that there are two distinct matching steps to trace out. To that effect, we assume a large separation of scales,

$$\Lambda_{\text{QCD}} \ll v \,, \qquad (4.13)$$

so that the Higgsing occurs first. The idea is that in this first Higgsing step we integrate out the heavy degrees of freedom (a heavy gauge field and a radial scalar mode) to get an intermediate effective description of the remaining light scalars coupled to quarks. Then we follow the RG flow through the subsequent chiral symmetry breaking transition by which quarks and gluons give way to pion degrees of freedom. We assume that, triggered by the flow to strong coupling at the low scale, chiral symmetry breaking and confinement occurs in the QCD sector more-or-less unaffected by the weak coupling to the dark sector that is mediated by the abelian gauge field.

### 4.3.1 Derivation of the mixed WZW term take 1: local form

To get a feel for how the mixed WZW term arises, we shall first work locally to derive the local approximation to the mixed WZW term given in Eq. (2.6), using the

gauge fixing procedure familiar from the electroweak theory.[11] In unitary gauge, and taking the unbroken $U(1)_\phi$ subgroup to be generated by $\sigma^3/2 \in \mathfrak{su}(2)_\phi$, we expand the doublet of complex scalar fields $\phi(x) := (\phi_1(x), \phi_2(x))^T$ around a minimum of the potential as

$$\phi(x) = e^{\frac{i}{2f_D}\chi^i(x)\sigma^i} \begin{pmatrix} 0 \\ v + \frac{\rho(x)}{\sqrt{2}} \end{pmatrix}, \tag{4.14}$$

where here the index $i$ runs only from 1 to 2, *i.e.* over the broken generators. The $\chi^3$ Goldstone mode has, in this gauge, been eaten to become the longitudinal mode of the Higgsed abelian gauge field $b$. For simplicity, we further assume a limit $m^2 \gg v^2$ (by taking $\lambda \gg 1$) to decouple the radial mode $\rho$, which we henceforth neglect, to obtain a non-linear sigma model description of the scalar sector in this Higgsed phase.

We now consider the effective field theory valid at energies $E$ in the intermediate régime,

$$\Lambda_{\text{QCD}} \ll E \ll v. \tag{4.15}$$

This means we can integrate out the heavy gauge field $b$, whose mass is order $v$ (assuming an order-1 gauge coupling). We do so at tree-level by setting $b$ to its classical equations of motion, and we work to leading order in the derivative expansion $\partial_\mu/v \sim E/v$, which is here equivalent to neglecting the kinetic term for $b$ in the equation of motion.

Then, the relevant terms in the Lagrangian that involve $b$ come from the kinetic terms for $\phi_i$ and $\Psi_i$, which read

$$\mathcal{L}_{\text{UV}} \supset b_\mu j_q^\mu + |(\partial - ib)\phi_i|^2 \supset v^2 b^2 + b_\mu j_\phi^\mu + b_\mu j_q^\mu, \tag{4.16}$$

where the quark and scalar currents are given locally by the 1-forms

$$j_q = \overline{\Psi}_i \gamma_\mu \Psi_i \, dx^\mu, \tag{4.17}$$

$$j_\phi = \frac{v^2}{2f_D^2} \epsilon_{ij} \chi^i d\chi^j, \qquad i, j = 1, 2. \tag{4.18}$$

We emphasize that the latter equation is derived under the assumption that we're in a coordinate patch in the vicinity of the origin $\chi^1 = \chi^2 = 0$, and does not necessarily hold away from such a patch. The leading order equations of motion then give

$$b = -\frac{1}{2v^2}(j_\phi + j_q) + \dots, \tag{4.19}$$

so after integrating out $b$ we generate effective mass dimension-6 operators of the form

$$\mathcal{L}_{\text{EFT}} \supset -\frac{1}{4v^2} (j_\phi + j_q)_\mu (j_\phi + j_q)^\mu, \tag{4.20}$$

---

[11]As anticipated above, the scalar QED part of our Lagrangian corresponds precisely to the electroweak theory describing the complex Higgs doublet, but with the $SU(2)_L$ gauge coupling turned off.

which is the leading order (and local) result of our intermediate EFT matching step. The cross-term will eventually match onto our mixed WZW term.

Now consider flowing further into the deep IR, *i.e.* to energy scales

$$E \ll \Lambda_{\text{QCD}} \,. \tag{4.21}$$

The QCD chiral symmetry $\text{SU}(N)_L \times \text{SU}(N)_R$ breaks spontaneously down to its diagonal subgroup $\text{SU}(N)_V$ due to the non-vanishing chiral condensate. The QCD part of the resulting sigma model description is known as the chiral Lagrangian, for which the leading order action is

$$S_\chi[g] = \int_{\Sigma_4} \frac{f_\pi^2}{4} \text{Tr} \, \left( \partial_\mu g^\dagger \partial^\mu g \right) + n_c \Gamma[g] \,, \tag{4.22}$$

where the dynamical field

$$g(x) = \exp\left( \frac{2i}{f_\pi} \pi_a(x) t_a \right) \in \text{SU}(N) \cong \frac{\text{SU}(N)_L \times \text{SU}(N)_R}{\text{SU}(N)_V} \tag{4.23}$$

describes the pions $\pi(x)$, with the pion decay constant $f_\pi$ and where $t^a$ are the generators of $\text{SU}(N)$. We include the usual WZW term $in_c\Gamma[g]$, with

$$\Gamma[g] = \frac{1}{240\pi^2} \int_{\Sigma_5} \text{Tr} \, \left( g^{-1}dg \right)^5 \,, \quad \partial\Sigma_5 = \Sigma_4 \,, \tag{4.24}$$

which is needed to match the 't Hooft anomalies in the $\text{SU}(N)^2$ QCD flavour symmetry.

Most important for us, however, is what becomes of the interaction term between the quark and the scalar sectors, which is the cross-term in the effective coupling in (4.20), namely

$$\mathcal{L}_{\text{int}} = -\frac{1}{2v^2} j_{q,\mu} j_\phi^\mu \,. \tag{4.25}$$

Due to confinement, the quarks now combine into baryons which carry $\text{U}(1)_q$ charge. Because one baryon consists of $n_c$ quarks, the baryon current $j_B$ is given in terms of the quark current $j_q$ by

$$j_q = n_c j_B \,, \tag{4.26}$$

with, importantly, a relative factor of $n_c$ appearing in the normalisation of these currents. In the chiral Lagrangian, the baryons can be identified as solitons (à la Skyrme [18]) formed from the pion fields. In this description, the baryon number current $j_B$ is then given in terms of $g$ by the topologically conserved form [19, 20]

$$\star j_B = \frac{1}{24\pi^2} \text{Tr} \, \left( g^{-1}dg \right)^3 = \frac{1}{24\pi^2} \frac{2}{f_\pi^3} f_{abc} d\pi_a \wedge d\pi_b \wedge d\pi_c + \mathcal{O}(\pi^4) \,, \tag{4.27}$$

and the integral of $\star j_B$ measures the baryon number of a pion field configuration.

The cross interaction term in our EFT Lagrangian becomes

$$\mathcal{L}_{\text{int}} = -\frac{n_c}{2v^2} j_{B,\mu} j_\phi^\mu. \tag{4.28}$$

Expanding both currents in terms of the pion fields $\pi_a(x)$ and the sigma model fields $\chi_i(x)$, and integrating by parts to move a derivative, we obtain the local Lagrangian

$$\mathcal{L}_{\text{int}} = \frac{n_c \epsilon^{\mu\nu\rho\sigma}}{48\pi^2 f_D^2 f_\pi^3} f_{abc} \epsilon_{ij} \pi_a \partial_\mu \pi_b \partial_\nu \pi_c \partial_\rho \chi_i \partial_\sigma \chi_j + \mathcal{O}(\pi^4\chi^2, \pi^3\chi^3). \tag{4.29}$$

This matches the local form of the mixed WZW term given in Eq. (2.6), with the IR coefficient fixed to be the number of colours in the QCD sector, $n = n_c$; precisely as the 2-group symmetry matching argument at the end of §4.1 suggested.

However, the naïve leading order EFT matching we have just demonstrated is not quite sufficient in this scenario, precisely because of the fundamentally topological nature of this interaction. By working with only the local form of the currents and Lagrangian terms (enforced by our use of local coordinates $\{\chi^i\}$), which we did to elucidate the perturbative physics as clearly as possible, we have ignored important and non-trivial topological data concerning the vacuum manifold $S^2$. In particular, one could not with this formalism hope to show that the global form of the topological term must be written in terms of $\text{Vol}_{S^2}$, and that the term therefore requires an extension to an auxiliary 5d bulk. In the following Subsection we will patch up our derivation, to give a globally-valid account of the EFT matching onto this mixed WZW term.

### 4.3.2 Derivation of the mixed WZW term take 2: global form

To make the topological information about the vacuum manifold (in particular, its non-trivial second homology and second homotopy groups, and associated winding number) manifest in the final mixed WZW, we have to derive it with a less direct implementation of gauge fixing, that does not require a local expansion of the underlying scalars $\phi_i$ in terms of the dark pion fields $\chi^i$.

First, instead of the unitary gauge used in Eq. (4.14), let us now expand the scalar fields around the vacuum manifold as

$$\phi_i = z_i + h_i, \qquad |z_1|^2 + |z_2|^2 = v^2, \tag{4.30}$$

where $z_1, z_2$ (subject to the constraint) describe the vacuum manifold $S^3$ of radius $v$, and $h_i$ are the transverse fluctuations that give rise to the single radial mode $\rho(x)$ after gauge fixing. As in the usual spontaneous symmetry breaking story, the potential $V(\phi)$ then tells us that the radial modes are massive and can be integrated out, while the vacuum manifold's degrees of freedom are massless, corresponding to the NGBs.

The effect of gauging the U(1) subgroup is that the affine coordinates $(z_1, z_2)$ become a pair of homogeneous coordinates $[z_1 : z_2]$, where we identify[12]

$$(z_1, z_2) \sim e^{i\alpha}(z_1, z_2) \tag{4.31}$$

These homogeneous coordinates form a familiar description of the 1-dimensional complex projective space $\mathbb{C}P^1$, which is topologically a 2-sphere $S^2$. We have now transitioned from a geometric description of spontaneous symmetry breaking (of a global symmetry) to a geometric description of the Higgs mechanism, where one would-be NGB morphs into the longitudinal mode of the gauge field and gets integrated out. The scalar electrodynamics sector is then effectively described by a sigma model whose target space is properly identified (globally) with the manifold $\mathbb{C}P^1$, which we will informally call the $\mathbb{C}P^1$-model.

We can then repeat the steps in the previous Subsection and integrate out massive fields to obtain the EFT

$$\mathcal{L}_{\text{EFT}} \supset -\frac{1}{4v^2}\left(j_\phi + j_q\right) \wedge \star \left(j_\phi + j_q\right) \tag{4.32}$$

at an intermediate energy scale between $v$ and $\Lambda_{\text{QCD}}$. The only difference here is the form $j_\phi$ takes. In the current gauge fixing scheme, we appear to have

$$j_\phi = -i\left(dz_i^* z_i - z_i^* dz_i\right) . \tag{4.33}$$

Again, flowing further down the RG flow replaces the QCD quark-gluon description with the chiral Lagrangian, with the quark current $j_q$ replaced by $n_c j_B$. Just like in the previous Subsection, one might be tempted to write the cross interaction term as

$$S_{\text{int}} \overset{?}{=} -\frac{n_c}{2v^2}\int_{\Sigma_4} \star j_B \wedge j_\phi\,, \tag{4.34}$$

but that would be wrong!

The problem with the proposed interaction (4.34) lies in the fact that it is not gauge invariant. This is because on $S^2$, the object

$$\mathcal{A}_\phi := \frac{1}{2v^2} j_\phi \tag{4.35}$$

is not a globally-defined form, but rather behaves like a gauge connection. This becomes evident in our new description via homogeneous coordinates $\{z_i\}$, which still suffer from a gauge redundancy. To wit, we observe that under the U(1) gauge transformation

$$(z_1, z_2) \to e^{i\alpha}(z_1, z_2), \qquad \Psi \to e^{i\alpha}\Psi \tag{4.36}$$

---

[12]We must also identify $\Psi \sim e^{i\alpha}\Psi$ at the same time since the quarks are also charged under this U(1) gauge symmetry.

that defines the homogeneous coordinates $[z_1 : z_2]$ on $S^2$, the connection $\mathcal{A}_\phi$ obtained from (4.33) transforms as

$$\mathcal{A}_\phi \to \mathcal{A}_\phi - d\alpha \,. \tag{4.37}$$

To emphasize the difference with our previous (necessarily local) description, the new description via homogeneous coordinates $\{z_i\}$ naturally covers the whole target space, but at the expense of having not fully fixed the gauge yet. And indeed one finds that the current $j_\phi$ is not gauge-invariant, but transforms (after an appropriate rescaling) as a connection. In contrast, things were fully gauged-fixed in the $\chi^i$-based formulae, but were necessarily restricted to a local patch near the origin, so not well-suited to studying field configurations that wind the $S^2$.

Fortunately, there is a known path to proceed in such a situation, because the putative Lagrangian (4.34) behaves exactly like a Chern–Simons term, which is similarly not gauge-invariant but which (iff properly-quantized) we know how to define rigorously in terms of the field strength by going to one dimension higher.[13] The correct interaction term takes the form

$$S_{\text{int}} = -n_c \int_{\Sigma_5} d\left(\star j_B \wedge \mathcal{A}_\phi\right) \tag{4.38}$$

$$= -n_c \int_{\Sigma_5} \star j_B \wedge d\mathcal{A}_\phi \tag{4.39}$$

where we have used the fact that $j_B$ is a topological current, that is $d \star j_B = 0$ off-shell.

All that remains is to show that $d\mathcal{A}_\phi$, which really denotes the curvature of the 1-form connection $\mathcal{A}_\phi$, is proportional to the volume form on the vacuum manifold $S^2$. Recall that this vacuum manifold is described by the pair of homogeneous coordinates $[z_1 : z_2]$ satisfying $|z_1|^2 + |z_2|^2 = v^2$. It can be shown (see, for instance, the book of Bott and Tu [43, §17]) that the volume form $\text{Vol}_{S^2}$ on this manifold, normalised so that $\int_{S^2} \text{Vol}_{S^2} = 1$, can be written in terms of $z_0, z_1$ as

$$\text{Vol}_{S^2} = -\frac{i}{2\pi v^2} dz_i^* \wedge dz_i \,. \tag{4.40}$$

On the other hand, we also find from Eq. (4.33) that $dj_\phi = 2i dz_i^* \wedge dz_i$. So, as promised, we obtain

$$d\mathcal{A}_\phi = -2\pi \text{Vol}_{S^2} \,. \tag{4.41}$$

---

[13]The exponentiated action here can also be defined as an invariant differential character [38–40] on the pion target space, the curvature of which is the globally-defined closed 5-form that we integrate in Eq. (4.43). This curvature form is analogous to the anomaly polynomial $\Phi_{d+2}$ in defining the Chern–Simons action in $d + 1$ dimensions. From this perspective, the action requires quantized coefficient precisely because it cannot be expressed via a locally-defined 4-form, and can be seen without passing to an extra dimension by instead patching together locally defined forms using the tools of Čech cohomology [22, 41, 42], and demanding consistency.

Note that this is consistent with $\int_{S^2} \mathrm{Vol}_{S^2} = 1$ although $\mathrm{Vol}_{S^2}$ appears to be an exact form, because $\mathcal{A}_\phi$ is really a connection 1-form (not a globally-defined 1-form) and $d\mathcal{A}_\phi$ is shorthand for its curvature. Note also that the first Chern number associated to this connection is correctly quantized, *viz.* $c_1 = \int_{X_2} \frac{d\mathcal{A}_\phi}{2\pi} \in \mathbb{Z}$ for $X_2$ any 2-cycle in $\Sigma_4$, and where we use the same notation $d\mathcal{A}_\phi$ for the pullback of $d\mathcal{A}_\phi$ under $z_i : \Sigma_4 \to S^2$.

After we replace $d\mathcal{A}_\phi$ in terms of $\mathrm{Vol}_{S^2}$, the cross interaction term now reads

$$S_{\mathrm{int}} = 2\pi n_c \int_{\Sigma_5} \star j_B \wedge \mathrm{Vol}_{S^2} \tag{4.42}$$

$$= 2\pi n_c \int_{\Sigma_5} \frac{1}{24\pi^2} \mathrm{Tr} \left( g^{-1} dg \right)^3 \wedge \mathrm{Vol}_{S^2}, \tag{4.43}$$

reproducing precisely (and globally) the mixed WZW term that we are after, with all the topological data manifest. Again, we see that the WZW coefficient is given by

$$n = n_c$$

More generally, if in the UV the abelian gauge field couples to the quarks with charge $X_q$ (but still charge $+1$ to the scalars), then the mixed WZW coefficient $n$ becomes $X_q n_c$. In this case, the Postnikov class appearing in the 2-group structure is also modified to $X_q n_c$ (as can be seen by straightforwardly adapting the argument of §4.1). The case of non-minimal scalar charge $X_\phi$ is slightly different, as we discuss in §5.

**A tree-level exact result**

We pause to make some further comments before continuing. First, we emphasize that it is extremely non-generic that we were able to match an interaction involving QCD through the chiral symmetry breaking transition into the chiral Lagrangian! This was only possible because the interaction with QCD was via baryon number current, which is robustly identified with a topologically conserved current in the IR. Likewise on the dark side, the coupling of the abelian gauge field is special, in that the connection 1-form associated to this coupling is a topologically non-trivial connection. When combined, these two special features contrive to mean that one obtains a *bona fide* quantized topological term — from integrating out a weakly coupled abelian gauge field at tree-level.

Due to the integer quantization of this coefficient (in appropriate units, which absorb the factors of $f_\pi$ and $v$ if we are using the local coordinate expressions), it follows that this tree-level matching result for the coefficient ought to be exact. The operator can only have an integer coefficient for consistency (as can be inferred purely from the low-energy EFT), and any corrections to this leading term, in the form of a perturbative series in the $\mathbb{R}$-valued couplings, could not maintain this

integrality as the couplings run under RG flow. This is exactly analogous to the non-renormalisation of the chiral anomaly. The difference is that for the anomaly the result is 1-loop exact, whereas here the leading order term is already there at tree-level.

This situation is reminiscent of how anomalies match for the Schwinger model (*i.e.* the 2d theory of a Dirac fermion coupled to a U(1) gauge field) not under RG flow but across the bosonization duality. In that case, the mixed anomaly between the vector and the axial U(1) symmetries on the fermionic side, which arise at 1-loop from the chiral fermion path integral, is matched by the mixed anomaly between the shift and winding symmetries on the bosonic side, which follows just from the tree-level equations of motion (see *e.g.* the lecture notes [44, §7.5.6]).

Another way to see this is that the coefficient $n$ appears in the 2-group current algebra where it is necessarily integer-quantized. It is this quantization that justi-fies the 'symmetry matching' across RG flows (akin to the more familiar 'anomaly matching') that we used in the beginning to suggest the no-go theorem of §3.

## 4.4  Phase structure of QCD coupled to scalar electrodynamics

In this Section, we briefly describe how the IR dynamics of QCD coupled to scalar electrodynamics, namely the UV theory described in §4.1, changes as we vary the dimensionless parameter

$$\mu^2 := \frac{m^2}{\Lambda_{\text{QCD}}^2} \tag{4.44}$$

while keeping the scalar quartic coupling $\lambda \sim \mathcal{O}(1)$ fixed, where recall $m^2$ is the mass-squared parameter for the scalars $\phi_i$ and $\Lambda_{\text{QCD}}$ is the strong coupling scale for the QCD sector at which the chiral symmetry breaking occurs.

(a). $\boldsymbol{\mu^2 \to +\infty}$. In this phase, the scalars are extremely massive and can be integrated out entirely. The remaining theory is QCD with gauge group $U(n_c)$ coupled to $N$ fundamental quarks. The gauged U(1) symmetry, which in the UV acted to rotate both scalars and quarks, in the IR acts only as a gauging of baryon number. There is no additional non-trivial U(1) global symmetry remaining once the scalars are lifted.

(b). $\boldsymbol{0 < \mu^2 < 1}$. In this case, chiral symmetry breaking occurs in the QCD sector while the scalars remain dynamical, and the photon (from gauging the mediator U(1)) remains massless. The IR theory consists of QCD mesons and baryons, with the baryons coupled weakly via the photon to two complex scalars of mass $m_\phi \sim \sqrt{\mu^2 \Lambda_{\text{QCD}}^2} < \Lambda_{\text{QCD}}$. Going to the deep IR, one would also integrate out the baryons and scalars and obtain a free theory of weakly-interacting massless mesons and photons.

(c). **$0 > \mu^2 > -1$**. Here the scalars condense, triggering the symmetry breaking
pattern described in the main text, but the QCD chiral symmetry breaking
transition occurs first. The deep IR phase is the same as that described in the
main text, namely of QCD and dark pions coupled to each other via the mixed
WZW term. But, if we assume a scale separation ($|\mu^2| \ll 1$), one can study the
EFT describing the intermediate phase. The situation is now 'reversed' to that
studied in §4.3, in that it features QCD pions and baryons, with the baryon
current coupled via the massive (but still dynamical) abelian gauge field to the
pair of dark complex scalars. In this intermediate phase, the 2-group structure
is matched by a term

$$S_{\text{int}} = 2\pi n_c \int_{M_5} \frac{\text{Tr } (g^{-1}dg)^3}{24\pi^2} \wedge \frac{h}{2\pi} \,, \tag{4.45}$$

where recall $h = db$ is the U(1) field strength. This ordering of the phase
transitions could just as well have been used to rigorously derive the emergence
of the mixed WZW term in the deep IR.

(d). **$\mu^2 \to -\infty$**. This is the limit discussed in the main text, which follows the RG
flow described in §4.3.

## 5 Variations of the scalar sector

In this Section we discuss three variations in the scalar sector of the theory.

### 5.1 Non-minimal scalar charge

In the main text (§4.3) we discussed the straightforward modification that follows
from varying the abelian quark charge $X_q$; in the UV, the term in the anomaly
polynomial responsible for the 2-group structure (and thus the Postnikov class) is
simply rescaled by $X_q$, and this tracks all the way through the RG matching to
rescale the coefficient of the WZW term.

But what if the scalar charge is taken to be non-minimal, $|X_\phi| \neq 1$? This naïvely
presents a puzzle, because the 2-group structure is not modified (depending only on
chiral fermion representations in the UV theory), but the coefficient of the WZW
*does* appear to be rescaled by $X_\phi$ upon matching, since the vertex coupling the gauge
field to the scalars is rescaled.

This requires a slightly more careful analysis, because the non-minimal scalar
charge alters the symmetry breaking structure induced by the scalars condensing.
Importantly, if the scalars have charge $|X_\phi| \neq 1$ under the gauge field (for which we
now take the quarks to have charge $+1$), the U(1) gauge symmetry is broken only
down to a discrete $\mathbb{Z}_{|X_\phi|} \subset$ U(1) by the condensate. The target space arising from

the global symmetry breaking in turn becomes the quotient

$$S^2/\mathbb{Z}_{|X_\phi|}\,, \tag{5.1}$$

for now ignoring the QCD part which is unchanged. This manifold is diffeomorphic to another 2-sphere (albeit with singular points appearing in the quotient metric at the poles), but the volume form must be rescaled,

$$\mathrm{Vol}_{S^2/\mathbb{Z}_{|X_\phi|}} = X_\phi \, \mathrm{Vol}_{S^2} \tag{5.2}$$

in order to remain integral (since we effectively integrate only over a wedge defined by opening angle $\Delta\phi = 2\pi/|X_\phi|$ to obtain the volume of the 'quotiented' sphere). The object $j_\phi$ obtained from the Lagrangian is clearly rescaled as $j_\phi \mapsto X_\phi j_\phi$, and so this factor drops out when we trade the curvature $dj_\phi$ for the volume form, that is $dj_\phi = -4\pi v^2 \mathrm{Vol}_{S^2}$ remains true.

The upshot is that the coefficient of the mixed WZW term does not depend on $X_\phi$ (not even on its sign). This is consistent with matching the 2-group symmetry, resolving our little puzzle.

## 5.2 From 2-sphere to 2-torus?

One might wonder if it is special that the two 'dark' pions live on $S^2$, or whether a similar story plays out for a non-linear sigma model on $\mathrm{SU}(N) \times K$, where $K$ is some other homogeneous space with $H^2_{\mathrm{dR}}(K) \neq 0$ so that there is a cohomologically non-trivial 2-form that can play the role of $\mathrm{Vol}_{S^2}$ in constructing the mixed WZW. In particular, one might consider the case $K = (\mathrm{U}(1) \times \mathrm{U}(1))/\{\cdot\} = T^2$, $i.e.$ with a pair of 'axion-like' dark pions living on a 2-torus. Let $(\phi_1, \phi_2) \in [0, 2\pi)^2$ denote coordinates on $T^2$ in this Subsection.

Even though a naïve cohomology-based classification of topological terms [45] would suggest there is a mixed WZW for this coset, there is in fact no such term if we insist the NLSM has exact $\mathrm{U}(1) \times \mathrm{U}(1)$ global symmetry acting by translations on $\phi_{1,2}$.[14] (In contrast, the WZW term on $\mathrm{SU}(N) \times (\mathrm{SU}(2)_\phi/\mathrm{U}(1)_\phi)$ studied above $is$ invariant under exact $\mathrm{SU}(N)_L \times \mathrm{SU}(N)_R \times \mathrm{SU}(2)_\phi$.) A putative WZW term, which one might define precisely to be a differential character with non-vanishing curvature form $\omega_{d+1}$, is $G$-invariant iff $\omega_{d+1}$ satisfies the so-called 'Manton condition' [22, 40], which requires the contraction of $\omega_{d+1}$ with each vector field generating the $G$-action (we assume $G$ is connected) be an exact form (not just closed). The putative 5-form $\omega \sim \mathrm{Tr}\,(g^{-1}dg)^3 d\phi_1 d\phi_2$ on $\mathrm{SU}(N) \times T^2$ violates this condition, because

$$\{\iota_{\partial_{\phi_1}}\omega, \iota_{\partial_{\phi_2}}\omega\} \sim \{\mathrm{Tr}\,(g^{-1}dg)^3 d\phi_2, -\mathrm{Tr}\,(g^{-1}dg)^3 d\phi_1\} \tag{5.3}$$

---

[14]We have in mind that the pions arise as Goldstones on $G/H$ following spontaneous symmetry breaking, starting from a $G$-invariant Lagrangian, and seek to construct the most general EFT consistent with symmetry. Alternatively, one can dispense with the global symmetry and view the scalar field theory as arising from a general non-linear sigma model, in which case there is no obstruction to defining the mixed WZW on $\mathrm{SU}(N) \times T^2$.

are closed but not exact 4-forms. A classification of *invariant* topological actions using invariant (differential) cohomology [40] tells us there is no such term in the IR consistent with the global $U(1)^2$ symmetry.[15]

We can also see the pathology from the point of view of 2-group symmetry. While naïvely there is still a locally conserved 2-form, associated to the closed volume form $\mathrm{Vol}_{T^2} = \frac{1}{4\pi^2} d\phi_1 \wedge d\phi_2$, there is in fact no topological charge associated to the putative 1-form symmetry because $\pi_2(S^1 \times S^1) = 0$, so there are no linking surfaces that are topologically 2-spheres through which to measure a monopole flux. So, there should be no line operators transforming non-trivially under this 1-form symmetry, with which to close the 2-group symmetry structure.

Of course, if we do not restrict our EFT to building invariants then there is nothing to prevent one from coupling QCD-like pions via the mixed WZW term to a pair of necessarily *pseudo* NGBs $\phi_{1,2}$ living on $T^2$.[16] The mixed WZW-like coupling just described provides a source of explicit $U(1)^2$ symmetry breaking, closely analogous to symmetry breaking by an ABJ anomaly, that contributes to the non-zero masses of the pNGBs. Under an axion shift symmetry $\phi_1 \to \phi_1 + \lambda_1$, the failure of the Manton condition encoded by (5.3) implies the mixed WZW action would shift by

$$ S \mapsto S + \int_{\Sigma_4} \frac{\lambda_1}{2\pi} \frac{n}{24\pi^2} \mathrm{Tr} \ (g^{-1}dg)^3 \frac{d\phi_2}{2\pi} \, , \tag{5.4}$$

which mimics the non-invariance due to an ABJ anomaly but with the instanton density $F \wedge F$ replaced by the 4-form $\propto \mathrm{Tr} \ (g^{-1}dg)^3 d\phi_i$.

## 5.3 From $\mathbb{C}P^1$ to $\mathbb{C}P^n$

Having discussed the torus and its subtleties, one might then ask if there are other cosets $K$, generalising $SU(2)/U(1)$, for which the mechanisms we have described do go through.

We remark that one set of examples is readily furnished by directly generalising the $\mathbb{C}P^1 \cong SU(2)/U(1) \cong S^2$ model to

$$ \mathbb{C}P^n \cong \frac{SU(n+1)}{S[U(1) \times U(n)]} \tag{5.5}$$

for $n \in \mathbb{Z}_{>1}$. The de Rham cohomology is

$$ H_{\mathrm{dR}}^\bullet(\mathbb{C}P^n) = \begin{cases} \mathbb{R} & n \text{ even} \, , 0 \le n \le n \\ 0 & n \text{ odd} \, , \end{cases} \tag{5.6}$$

---

[15]This failure of invariance is a higher-dimensional avatar of the fact that coupling a quantum particle on a torus to a homogeneous (classically) translationally-invariant magnetic field breaks translations down to a discrete subgroup, a fact noticed long ago by Manton [46]. A similar phenomenon occurs [47] for a non-minimal composite Higgs model proposed in [48].

[16]With non-zero masses, a phenomenologist would refer to such pNGBs on $T^2$ as a pair of 'axion-like particles', or ALPs.

and because $\mathbb{C}P^n$ is a symmetric space, the invariant forms are in 1-to-1 with cohomology classes. Moreover, because $G = \mathrm{SU}(n+1)$ is simple, the Manton condition discussed above reduces simply to requiring $G$-invariance of the differential form. Thus, a representative form $\Omega$ for the generator of the cohomology ring above, which can be identified with the Kähler form, can be used to construct a mixed WZW action of the form

$$S[\Sigma_4] = \int_{X_5} \frac{m}{24\pi^2} \mathrm{Tr}\ (g^{-1}dg)^3 \wedge \Omega\,, \qquad n \in \mathbb{Z},\ \partial X_5 = \Sigma_4\,. \qquad (5.7)$$

In homogeneous coordinates that generalise those introduced for $S^2 \cong \mathbb{C}P^1$ in §4.3, defined as

$$\left\{ z_i \in \mathbb{C}^{n+1}\backslash\{0\} \quad | \quad \sum_i |z_i|^2 = v^2, \quad z_i \sim e^{i\alpha}z_i\ \forall \alpha \in \mathbb{R}/2\pi\mathbb{Z} \right\} \qquad i = 0,\ldots n\,, \qquad (5.8)$$

we can choose a representative 2-form to be given by the Kähler form (see *e.g.* [49])

$$\Omega \sim -\frac{i}{v^2} \sum_{i=0}^{n} dz_i \wedge dz_i^*\,, \qquad (5.9)$$

appropriately normalised to have integer periods. This term, for non-zero coefficient $m$, encodes non-trivial 2-group symmetry exactly as for the $n = 1$ case studied at length in this paper.

The UV completion via QCD coupled to scalar electrodynamics should also generalise directly from $\mathbb{C}P^1$ to $\mathbb{C}P^n$, by passing from 2 complex scalars $\phi_i$ to $n+1$ complex scalars, all coupled with the same charge to the abelian gauge field $b$ (that also couples universally to quarks in a vector-like fashion as before). This many-scalar model could provide an interesting variant of the dark matter portal mechanism proposed in [14].

Yet further generalisation is possible if we allow $K$ to be a more general manifold $M$, not necessarily a coset arising from spontaneous symmetry breaking. The same mixed WZW term exists whenever $H_{\mathrm{dR}}^2(M)$ is non-trivial, with $\Omega$ similarly picked to be a 2-form representative of a non-trivial element in $H_{\mathrm{dR}}^2(M)$ with unit period. A partial UV-completion is provided by QCD coupled to a non-linear sigma model with target space being the line bundle $L \to M$ over $M$ whose Chern class is given by $\Omega$. We then gauge the diagonal between the $\mathrm{U}(1)_q$ quark number symmetry and the $\mathrm{U}(1)$ symmetry acting on the fibre of $L$.[17] The crucial non-trivial 2-group structure then arises from the mixed 't Hooft anomaly between $\mathrm{U}(1)_q$ and $\mathrm{SU}(N)_{L/R}$ after this gauging [34] (the same phenomenon also happens when gauging finite groups with mixed 't Hooft anomalies [50]).

---

[17]We thank Y. Tachikawa for suggesting this more general approach.

# 6  Gauged version

In this Section, we discuss the case in which an anomaly-free U(1) subgroup of the QCD 0-form flavour symmetry $SU(N)_L \times SU(N)_R$ is gauged. This is a physically important scenario, allowing one to describe for instance the gauging of electromagnetism in our extension of QCD by the $S^2$ pions.

## 6.1  Non-invertible symmetry: a first look in the IR

To be concrete, let us for now take $N = 3$ flavour QCD, and consider gauging the vector-like $U(1)_Q \subset SU(3)_{L+R}$ generated by

$$Q = \begin{pmatrix} 2 & & \\ & -1 & \\ & & -1 \end{pmatrix} . \tag{6.1}$$

Let $f_Q = da$ denote the corresponding abelian field strength.[18] As alluded to above in §2.2 (and used in [14]), there is a term (amongst others) in the gauged mixed WZW action like

$$S_{\text{WZW}} \supset 2\pi n_c \int_{\Sigma_4} \frac{\pi_0}{2\pi f_\pi} \frac{f_Q}{2\pi} \wedge \text{Vol}_{S^2} . \tag{6.2}$$

This arises because $\text{Tr}\,[(t_L^3 - t_R^3), Q] \neq 0$, where recall $t_L^3 - t_R^3$ is the generator of the neutral pion shift symmetry, where we adopt the usual Gell-Mann basis for $\mathfrak{su}(3)$.

This term in the action now encodes not the 2-group global symmetry relation from before, but a genuine breaking of the global axial symmetry: doing a shift $\pi_0 \to \pi_0 + \alpha f_\pi$ gives

$$\delta_\alpha S_{\text{WZW}} = 2\pi n_c \int_{\Sigma_4} \frac{\alpha}{2\pi} \frac{f_Q}{2\pi} \wedge \text{Vol}_{S^2} , \tag{6.3}$$

which can be non-zero for instance when evaluated on a spacetime manifold with topology $\Sigma_4 = S^2 \times S^2$. This is analogous to the breaking of a global symmetry via an abelian ABJ anomaly — or, in modern parlance, a non-invertible symmetry [51, 52] — but with a mixed operator involving both $f_Q$ and the $S^2$ winding number appearing in the anomalous variation.

## 6.2  Non-invertible symmetry from the UV anomaly polynomial

This non-invertible symmetry structure can be traced up to our UV completion (§4) via QCD coupled to SED. There, as we pass to the UV, the winding number on $S^2$ becomes identified with the field strength for the U(1) gauge field $b$ from before (that couples to baryon number on the QCD side). In this UV theory, which now

---

[18]Note that we have redefined $a$ with respect to previous Sections to here denote the abelian gauge field for electromagnetism (rather than the QCD gluon field as before).

has two gauged U(1) factors that we call $\mathrm{U}(1)_Q$ and $\mathrm{U}(1)_b$ in what we hope is an obvious notation, there is an abelian ABJ anomaly between the global axial current generating the pion shift and the two different gauged U(1) groups. That is, a term in the anomaly polynomial $\propto F \wedge f_Q \wedge h$ is responsible for the anomalous shift (6.3) in the UV theory, where $F$ is the background field strength for the axial current $\mathrm{U}(1)_A$ under consideration, and $h = db$ still.

Let's see how this works more explicitly. The quantum numbers of the quarks under the various gauge groups and the $\mathrm{U}(1)_A$ chiral global symmetry generated by $t_L^3 - t_R^3$ are given in Table 1 below. Note that both the gauged U(1)s are vector-like

|  | $\mathrm{SU}(n_c)$ | $\mathrm{U}(1)_Q$ | $\mathrm{U}(1)_{\mathrm{SED}}$ | $\mathrm{U}(1)_A$ |
|---|---|---|---|---|
| $\psi_1$ | $\mathbf{n_c}$ | $+2$ | $+1$ | $+1$ |
| $\psi_2$ | $\mathbf{n_c}$ | $-1$ | $+1$ | $-1$ |
| $\psi_3$ | $\mathbf{n_c}$ | $-1$ | $+1$ | $0$ |
| $\tilde{\psi}_1$ | $\overline{\mathbf{n_c}}$ | $-2$ | $-1$ | $+1$ |
| $\tilde{\psi}_2$ | $\overline{\mathbf{n_c}}$ | $+1$ | $-1$ | $-1$ |
| $\tilde{\psi}_3$ | $\overline{\mathbf{n_c}}$ | $+1$ | $-1$ | $0$ |

**Table 1**. Quantum numbers of the quark fields under various U(1) symmetries relevant to the gauged version of our theory.

and so trivially free of gauge anomalies. But because of the chiral nature of the global symmetry $\mathrm{U}(1)_A$, it is possible that there may be anomalies proportional to the background gauge field $F$. Indeed, turning on the background field $F$ for $\mathrm{U}(1)_A$, the anomaly polynomial reads

$$\Phi_6(F) = \frac{3n_c}{(2\pi)^3} \left( F \wedge f_Q \wedge f_Q + 2F \wedge f_Q \wedge h \right) . \tag{6.4}$$

The first term on the right-hand-side encodes the usual ABJ anomaly between $\mathrm{U}(1)_A$ and $\mathrm{U}(1)_Q$ responsible for $\pi^0 \to \gamma\gamma$ decay in real-world QCD, while the second term encodes the new effect that is our main interest here.

With recent advances in our understanding of generalised symmetries, we know that a non-invertible symmetry emerges from the $\mathrm{U}(1)_A$ symmetry destroyed by this pair of anomalies [51, 52]. The symmetry defect will take the form

$$U_\beta = \exp\left( 2\pi i \beta \int_{M_3} \star J - i\frac{3n_c\beta}{2\pi} \int_{M_3} (a + 2b) \wedge f \right), \quad \beta \in [0, 1), \tag{6.5}$$

with $\beta$ being rational.[19] The ill-defined CS terms are shorthands for 3d topological quantum field theories (TQFTs) localised on the defect submanifold $M_3$ which couple to the U(1) fields. For example, if we take $\beta = 1/6n_cK$, the symmetry defect becomes

$$U_{1/6n_cK} = \exp\left( \frac{2\pi i}{6n_cK} \int_{M_3} \star J - \frac{i}{4\pi K} \int_{M_3} [(a + b) \wedge d(a + b) - b \wedge db] \right). \tag{6.6}$$

---

[19]For a slightly different take on the range of the transformation's parameter, see Refs. [53, 54].

The second term in the exponential consists of two fractional CS theories, which can be properly defined with help from two auxiliary dynamical[20] U(1) gauge fields $c_1$ and $c_2$, localised on $M_3$, via

$$i \int_{M_3} \left( \frac{K}{4\pi} c_1 dc_1 - \frac{K}{4\pi} c_2 dc_2 \right) + i \int_{M_3} \left( \frac{1}{2\pi} c_1 (da + db) - \frac{1}{2\pi} c_2 db \right) , \qquad (6.7)$$

The first term indicates that this TQFT is the $U(1)_K \times U(1)_{-K}$ CS theory, while the second term specifies the coupling between the auxiliary fields and the 4d U(1) fields $a$ and $b$. For other values of $\beta \in \mathbb{Q}$ that result in the CS coefficient being $p/K$ instead of the 'unit fraction' $1/K$, Eq. (6.7) can be generalised in terms of a certain minimal $\mathbb{Z}_K$ TQFT, usually denoted by $\mathcal{A}^{N,p}$ [51], which we will not elaborate on further here. Interested readers are invited to see Refs. [10, 55] for detailed study of this particular TQFT.

### 6.3   Full global structure: 2-group plus non-invertible symmetry

One can consider turning on background gauge fields for other global symmetry currents that remain after gauging $U(1)_Q$. To analyse the symmetry structure more generally, not just for the specific $U(1)_A$ subgroup in Table 1 that shifts the $\pi^0$, we first have to work out what the remaining flavour symmetry $G'_{\text{flavour}}$ is after gauging.

We claim that the faithful global symmetry becomes

$$G'_{\text{flavour}} \cong \frac{U(2)_L \times U(2)_R}{U(1)} . \qquad (6.8)$$

To see this, note first that we can still treat the left- and the right-handed quarks independently. Since the U(1) EM subgroup is diagonal, what happens to the left-handed component must be the same as what happens to the right-handed component. The maximal subgroup of $SU(3)_L$ that commutes with the U(1) subgroup of $SU(3)_L$ generated by $Q$ consists of matrices of the form

$$U = \begin{pmatrix} e^{i\phi} & 0 \\ 0 & V \end{pmatrix} \qquad (6.9)$$

where $V$ is a U(2) matrix, so it can be written as $V = e^{i\theta}\tilde{V}$ with $\tilde{V} \in SU(2)$. Since the matrix $U$ must still be unimodular, we need $\det U = 1$, or $e^{i\phi + 2i\theta} = 1$. This means we need $U$ to be of the form

$$U = \begin{pmatrix} e^{-2i\theta} & 0 \\ 0 & e^{i\theta}\tilde{V} \end{pmatrix} \qquad (6.10)$$

Matrices of this form form the group

$$S\left[U(1) \times SU(2)\right]_L \cong U(2) , \qquad (6.11)$$

---

[20]This means $c_1$ and $c_2$ must be path-integrated implicitly in the definition of $U_{1/6n_c K}$.

where the isomorphism follows from the fact that the top-left entry is entirely fixed by the U(2) matrix in the bottom-right. However, we should not be too quick to conclude that the remaining global symmetry is $\tilde{G}' = \mathrm{U}(2)_L \times \mathrm{U}(2)_R$. The catch comes from the fact that an element $u$ of the gauged $\mathrm{U}(1)_Q \subset \mathrm{SU}(3)_L \times \mathrm{SU}(3)_R$ subgroup takes the form

$$\left\{ u = \begin{pmatrix} e^{2i\alpha} & & \\ & e^{-i\alpha} & \\ & & e^{-i\alpha} \end{pmatrix}, \begin{pmatrix} e^{-2i\alpha} & & \\ & e^{i\alpha} & \\ & & e^{i\alpha} \end{pmatrix} \right\}, \tag{6.12}$$

which is in the centre of $\tilde{G}'$. Therefore, two elements of $\tilde{G}'$ related to each other by $u$ must be identified. This reduces the full symmetry from $\tilde{G}'$ down to the $G'_{\mathrm{flavour}}$ group written in Eq. (6.8), as claimed.

From here we can determine the symmetry/anomaly structure of this theory, now turning on background gauge fields for a general global symmetry current (rather than just for the particular $\mathrm{U}(1)_A$ choice, as in (6.4)). To do so, it is convenient to start from the anomaly polynomial $\Phi_6$ before gauging $\mathrm{U}(1)_Q$, namely

$$\Phi_6(F_L, F_R) = \frac{n_c}{3!} \frac{1}{(2\pi)^3} \left[ \mathrm{Tr}\, F_L^3 - \mathrm{Tr}\, F_R^3 \right] + \frac{n_c}{2} \frac{h}{2\pi} \left[ \mathrm{Tr}\, \left( \frac{F_L}{2\pi} \right)^2 - \mathrm{Tr}\, \left( \frac{F_R}{2\pi} \right)^2 \right], \tag{6.13}$$

and replace the non-abelian background gauge fields $F_L$ and $F_R$ by the combinations

$$F_L = f_Q Q + F'_L, \tag{6.14}$$
$$F_R = f_Q Q + F'_R, \tag{6.15}$$

where $F'_L$ and $F'_R$ are the background fields for the global symmetry that remains after gauging $\mathrm{U}(1)_Q$, which means they must commute with the gauge group, *i.e.* must satisfy

$$[Q, F'_L] = [Q, F'_R] = 0. \tag{6.16}$$

After making this replacement, the anomaly polynomial becomes

$$\Phi_6(F_L, F_R) = \Phi_6(F'_L, F'_R) + \frac{n_c}{(2\pi)^3} \left[ \frac{1}{2} f_Q^2 \mathrm{Tr}\, \left( Q^2(F'_L - F'_R) \right) + h f_Q \mathrm{Tr}\, \left( Q(F'_L - F'_R) \right) \right], \tag{6.17}$$

where wedge products between the various 2-form field strengths are implicit. Locally, we can write $F'_L$ and $F'_R$ as elements of the Lie algebra of $S[\mathrm{U}(1) \times \mathrm{U}(2)]_{L/R} \cong \mathrm{U}(2)_{L/R}$. Letting $\mathcal{F}_L \in \mathrm{U}(2)_L$, and $\mathcal{F}_R \in \mathrm{U}(2)_R$, we can write

$$F'_L = \begin{pmatrix} -\mathrm{Tr}\, \mathcal{F}_L & \\ & \mathcal{F}_L \end{pmatrix}, \qquad F'_R = \begin{pmatrix} -\mathrm{Tr}\, \mathcal{F}_R & \\ & \mathcal{F}_R \end{pmatrix}. \tag{6.18}$$

Substituting into (6.17) and evaluating the traces, the anomaly polynomial $\Phi_6$ reduces to

$$\Phi_6 = \Phi'_6 - \frac{3n_c}{(2\pi)^3} \left( \frac{f_Q^2}{2} + h f_Q \right) (\mathrm{Tr}\, \mathcal{F}_L - \mathrm{Tr}\, \mathcal{F}_R), \tag{6.19}$$

where we use $\Phi'_6 = \Phi_6(F'_L, F'_R)$ as a shorthand.

**Non-invertible symmetry part.** The last term of the anomaly polynomial indicates that axial generators in $U(2)_{L/R}$ with non-vanishing trace, which can in general be expressed as a linear combination of the pion shift generator picked out in (6.4) plus a $t^8_L - t^8_R$ component (that also shifts the $\eta$ meson), suffer a mixed abelian ABJ anomaly with $h \wedge f_Q$, becoming non-invertible in the process [51, 52]. As described above, this is in addition to the usual contribution $\propto f_Q^2$ that is already there for pure QCD with gauged electromagnetism.

**2-group symmetry part.** We next examine the part $\Phi'_6$ to show that there remains a subgroup of the flavour symmetry $G'_{\text{flavour}}$ that participates in a 2-group structure after our gauging of $U(1)_Q$, that links the 0-form flavour symmetry to the magnetic 1-form symmetry for $U(1)_b$. Using the fact that $\mathcal{F}_L$ and $\mathcal{F}_R$ are $U(2)$ field strengths, we can write them as

$$\mathcal{F}_{L/R} = \frac{1}{2}\text{Tr } \mathcal{F}_{L/R} \, \mathbf{1}_2 + \tilde{\mathcal{F}}_{L/R} \tag{6.20}$$

where $\tilde{\mathcal{F}}_{L/R}$ are $\mathfrak{su}(2)$-valued. The part of the anomaly polynomial denoted $\Phi'_6$ then reads

$$\Phi'_6 = \frac{n_c}{3!}\frac{\text{Tr } \mathcal{F}_L}{(2\pi)^3}\left[-\frac{3}{4}(\text{Tr } \mathcal{F}_L)^2 + \frac{3}{2}\text{Tr } (\tilde{\mathcal{F}}_L^2)\right] \tag{6.21}$$

$$+ \frac{n_c}{2}\frac{h}{(2\pi)^3}\left[\frac{3}{2}(\text{Tr } \mathcal{F}_L)^2 + \text{Tr } (\tilde{\mathcal{F}}_L^2)\right] - (L \leftrightarrow R) \,. \tag{6.22}$$

Let us digest the various terms appearing here:

- The top line in this expansion of $\Phi'_6$ captures the ordinary 't Hooft anomaly in $U(2)_{L/R}$.

- The second line partly tells us that there is still the 2-group structure between the 1-form symmetry and the $SU(2)_{L/R}$ part of the flavour symmetry with the same Postnikov classes as before. Focussing on the non-abelian part $\propto h \wedge \text{Tr } (\tilde{\mathcal{F}}_L^2)$, the interpretation of this 2-group global symmetry is the same as before. For instance, one can consider the axial transformation generated by

$$X_L = -X_R = \begin{pmatrix} 0 & & \\ & 1 & \\ & & -1 \end{pmatrix}, \tag{6.23}$$

which does not suffer from any ABJ-like anomaly because

$$\text{Tr } (XQ^2) = \text{Tr } (XQ) = 0 \tag{6.24}$$

and so defines a proper invertible symmetry. This $X$ (together with a whole $\mathfrak{su}(2)$ subalgebra in the lower-right block) participates in a 2-group current algebra with $h$, much like before but reduced from SU(3)s to SU(2)s.

- The abelian part $\propto h \wedge (\text{Tr } \mathcal{F}_L^2)$, however, introduces a qualitatively new kind of symmetry structure: there is a non-trivial 2-group relation between the magnetic 1-form symmetry and the abelian part of the remaining flavour symmetry which is itself non-invertible, as we previously explained.

To our knowledge, this kind of global symmetry (namely one which, in the old terminology, participates in both ABJ anomalies and operator-valued mixed anomalies with gauged currents) has not been studied before, and will be explored in future work.

# 7 Conclusions and Outlook

Topological WZW terms play a fundamental role in low-energy theories of pions that arise from confining gauge theories: they are needed to match chiral anomalies. Our starting point in this paper is a curious example of a pion EFT, on a coset $\text{SU}(N) \times S^2$, that features a WZW term *not* related to any underlying chiral anomaly. While this term defines a perfectly sensible-looking EFT, it is not *a priori* clear how this might arise from a microscopic theory, and why its coefficient should end up quantized if not mandated by anomaly matching.

We find that this WZW term encodes not an anomaly, but a generalised 2-group global symmetry structure (that mixes ordinary flavour symmetries with a 1-form symmetry). Like the anomaly, this 2-group structure is rigid, as manifest at low-energies in the quantization condition for the WZW term, and so can be used to check the consistency of possible UV completions. Through this notion of 'symmetry matching' (as opposed to anomaly matching), we rule out naïve QCD-like completions of this pion EFT, and instead propose a weakly-coupled completion involving a QCD sector and a $\mathbb{C}P^1$ sector, coupled weakly together in a very particular way by an abelian gauge field that gets Higgsed along the RG flow. Strikingly, by integrating out the weakly coupled gauge boson at tree-level one ends up with a quantized topological term – which moreover implies the tree-level matching is exact. Even though loop corrections are expected to vanish, a 'global' form of EFT matching (topologically-speaking) is nonetheless needed to match precisely, which is conveniently handled by using homogeneous coordinates on $\mathbb{C}P^1$. This RG flow is a highly non-generic one, and occurs because the abelian gauge field couples to topologically non-trivial currents in both the QCD and $\mathbb{C}P^1$ sectors.

We discussed several variations of the setup, for instance, several alternatives to the $\mathbb{C}P^1$ model on the scalar side. We furthermore examined the important scenario in which an anomaly-free subgroup of the QCD flavour symmetry (such as QED) is gauged. This gives rise to a more complicated generalised symmetry structure involving both 2-group and non-invertible symmetry. This also points to a new kind of symmetry structure, that we wish to study in future work, since the theory nec-

essarily yields global symmetry currents $F$ that participate simultaneously in mixed anomalies of both '$Fff$' and '$FFf$' type. In addition, we wish to study this phenomenon of symmetry matching by WZW terms in a more general context, including discrete and/or non-perturbative examples, and theories outside 3+1 dimensions.

Lastly, turning to phenomenology, we aim to apply these ideas to investigate UV completions of the topological portal to dark matter proposed in [14]. With an explicit (and weakly-coupled) model in hand, one can compute the full set of phenomenological predictions in both collider and cosmological observables, to characterise the viable parameter space and the best probes of this scenario.

## Acknowledgement

We thank Philip Boyle-Smith, Hitoshi Murayama, Yuji Tachikawa, David Tong, and Ethan Torres for stimulating discussions. We are further grateful to Yuji Tachikawa for detailed comments on the draft. JD thanks Admir Greljo and Nudžeim Selimović for collaborating on the related project [14]. NL is supported by the STFC consolidated grant in Particles, Strings and Cosmology number ST/T000708/1 and the Royal Society of London. We thank King's College London for hosting us for 1 productive week spent working on this project, and lastly we thank the Watch House Café, Somerset House, for providing us with excellent coffee in this time.

# A  Computation of $\tilde{\Omega}_4^{\mathbf{Spin}}\left(\mathbf{SU}(N) \times S^2\right)$

In this Appendix, we present our computation of the reduced spin bordism groups of $\mathrm{SU}(N) \times S^2$ up to degree 4, using the Adams spectral sequence (ASS) [25]. In particular, we will show that

$$\tilde{\Omega}_4^{\mathrm{Spin}}\left(\mathrm{SU}(N) \times S^2\right) \cong \mathbb{Z}_2\,. \tag{A.1}$$

A readable practice guide on how to use the ASS to compute bordism groups can be found in *e.g.* Refs. [56, 57] (see also Appendix A1 of Ref. [58] for a brief summary of the general method).

The ASS relevant for us is a cohomological spectral sequence, consisting of a sequence of bi-graded abelian groups $E_r^{s,t}$, $r = 2, 3, 4, \ldots$ with gradings $s, t \geq 0$, together with differentials $d_r : E_r^{s,t} \to E_r^{s+r,t-r+1}$ forming layers of cochain complexes. We refer to the set of entries for a fixed value of the index $r$ as a 'page' of the sequence. An element $E_r^{s,t}$ of a page is determined from the previous page by taking the homology with respect to the differentials:

$$E_{r+1}^{s,t} = \frac{\ker\left(d_r : E_r^{s,t} \to E_r^{s+r,t-r+1}\right)}{\mathrm{im}\left(d_r : E_r^{s-r,t+r-1} \to E_r^{s,t}\right)} \tag{A.2}$$

For our purpose, the initial $E_2$ page of the ASS, and what it converges to, is given by

$$E_2^{s,t} = \text{Ext}_{\mathcal{A}(1)}^{s,t} \left( \tilde{H}^\bullet(\text{SU}(N) \times S^2; \mathbb{Z}_2), \mathbb{Z}_2 \right) \implies \left( \tilde{\Omega}_{t-s}^{\text{Spin}} \left( \text{SU}(N) \times S^2 \right) \right)_2^\wedge, \quad \text{(A.3)}$$

where $\mathcal{A}(1)$ is the Steenrod subalgebra spanned by the Steenrod squares operations $\{1, \text{Sq}^1, \text{Sq}^2\}$, and $(\cdot)_2^\wedge$ denotes the 2-completion. Strictly speaking, this ASS will only give us information about the free and the 2-torsion parts of the bordism groups. In our case, however, we can check by other means, such as the Atiyah–Hirzebruch spectral sequence (AHSS), that there are no other torsions present.[21] We therefore obtain complete information regarding the bordism groups we want to compute using this method – provided we can solve the spectral sequence.

To start, we need to know the mod 2 cohomology ring of $\text{SU}(N) \times S^2$ as an $\mathcal{A}(1)$-module. The mod 2 cohomology ring of $\text{SU}(N)$ is given by [59]

$$H^\bullet(\text{SU}(N); \mathbb{Z}_2) \cong \bigwedge\nolimits_{\mathbb{Z}_2} [x_3, x_5, \dots, x_{2N-1}], \quad \text{(A.4)}$$

that is, it is the exterior algebra on generators $x_3, x_5, \dots, x_{2N-1}$ with integer mod 2 coefficients, where $x_i$ are generators in degree $i$. The $\mathcal{A}(1)$-module structure is given by the action of the Steenrod squares on the generators:

$$\text{Sq}^1 x_i = 0, \qquad \text{Sq}^2 x_{2j-1} = \binom{j-1}{i} x_{2i+2j-1}. \quad \text{(A.5)}$$

Similarly, we can write the mod 2 cohomology ring of $S^2$ as

$$H^\bullet(S^2; \mathbb{Z}_2) \cong \bigwedge\nolimits_{\mathbb{Z}_2} [x_2], \quad \text{(A.6)}$$

with $\text{Sq}^1 x_2 = \text{Sq}^2 x_2 = 0$ on dimensional grounds. Applying Künneth's theorem, we obtain

$$H^\bullet(\text{SU}(N) \times S^2; \mathbb{Z}_2) \cong \bigwedge\nolimits_{\mathbb{Z}_2} [x_2, x_3, x_5, \dots, x_{2N-1}]. \quad \text{(A.7)}$$

The graphical representation of its reduced version (*i.e.* that obtained by ignoring the degree 0 generator) as an $\mathcal{A}(1)$-module is shown in Fig. 1, up to degree 5. Straight lines (of which there happen to be none) represent the action of $\text{Sq}^1$, while the curved lines represent the action of $\text{Sq}^2$.

We are now ready to compute the $E_2$ page of the spectral sequence, which is given by $E_2^{s,t} = \text{Ext}_{\mathcal{A}(1)}^{s,t} (H^\bullet(\text{SU}(N) \times S^2), \mathbb{Z}_2)$. As this is a direct sum of the functor $\text{Ext}_{\mathcal{A}(1)}^{s,t}(-, \mathbb{Z}_2)$ applied on each connected component in Fig. 1, which are known [56], we simply stitch them together and present the result graphically in Fig. 2 in what is known as the 'Adams chart' for $\text{Ext}_{\mathcal{A}(1)}^{s,t} (H^\bullet(\text{SU}(N) \times S^2), \mathbb{Z}_2)$.

---

[21]We find the AHSS on its own, however, is not sufficient to deduce the bordism group (A.1) due to unknown differentials.

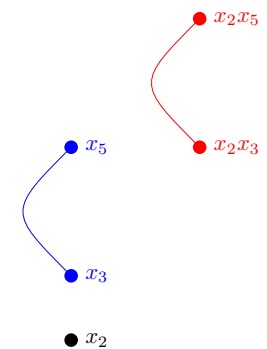

**Figure 1.** $\tilde{H}^\bullet(\mathrm{SU}(N) \times S^2)$ as an $\mathcal{A}(1)$-module.

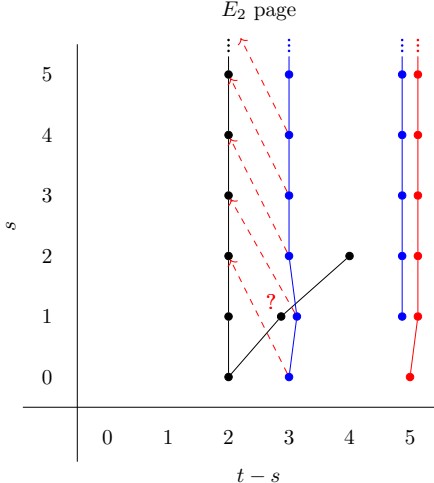

**Figure 2.** The Adams chart for $\mathrm{Ext}_{\mathcal{A}(1)}^{s,t}\left(H^\bullet(\mathrm{SU}(N) \times S^2), \mathbb{Z}_2\right)$.

We next solve the spectral sequence by 'turning the pages', taking successive homology with respect to the differentials to go from one $E_r$ page to the next. This is done until no elements in the range we are interested in change any more, at which point we say that the sequence stabilises. On the $E_2$ page, the only possible non-trivial differentials $d_2 : E_2^{s,t} \to E_2^{s+2,t+1}$ in the range of $t - s$ that we are interested in are from the column $t - s = 3$ to the column $t - s = 2$, as indicated in Fig. 2. By a comparison with the AHSS, we can easily determine that these differentials, as well as similar differentials on subsequent pages, are trivial. The spectral sequence in the relevant range $t - s \leq 4$ therefore stabilises already on this page. We can then read off the reduced bordism groups directly from the Adams chart, shown in Table 2 below. In particular, the reduced spin bordism group of $\mathrm{SU}(N) \times S^2$ in degree 4 is $\mathbb{Z}_2$ as claimed.

| $i$ | 0 | 1 | 2 | 3 | 4 |
|---|---|---|---|---|---|
| $\tilde{\Omega}_i^{\mathrm{Spin}}(\mathrm{SU}(N) \times S^2)$ | 0 | 0 | $\mathbb{Z}$ | $\mathbb{Z} \oplus \mathbb{Z}_2$ | $\mathbb{Z}_2$ |

**Table 2**. The reduced spin bordism groups of $\mathrm{SU}(N) \times S^2$ up to degree 4.

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
