# Peer review of "WZW terms without anomalies: generalised symmetries in chiral Lagrangians"

_SciPost Physics_

## Round 1 · Referee Report · Anonymous (Referee 1) · 2024-9-24

Strengths

- Clear and explicit explanation
- New type of symmetry matching between gauge theory and sigma model

Report

This manuscript studies the symmetry of a σ-model with a particular topological term and its possible UV completion. Specifically, the topological term is specified by a (generalized) cohomology class of the target manifold SU(N)×S2, which can be seen as a generalization of the Wess–Zumino–Witten (WZW) term. The authors argue that the σ-model possesses a continuous 2-group symmetry that enforces the presence of a continuous 1-form symmetry, which in turn prohibits strongly-coupled non-abelian gauge theory as its UV completion. The authors propose a UV completion involving an abelian gauge field that matches the symmetry structure.

The manuscript is well-written, and the argument is clear. The results are interesting, and as the authors claim, they can potentially be applied to phenomenological problems. Therefore, I recommend the publication of this manuscript after the following minor issues are addressed:

Requested changes

1. The manuscript suggests a theory with abelian gauge field as a UV completion of the σ-model. However, an Abelian gauge theory is not UV complete due to the Landau pole. Could authors provide a comment on how this "UV" abelian gauge theory could be further UV completed. In particular, does the no-go theorem imply that the flavor symmetry must be broken at some scale?

2. On Page 5: "(Co)homological [22] (or (co)bordism-based [23, 24]) classifications of Wess–Zumino–Witten (WZW) terms." I believe the (generalized) cohomological perspective of WZW terms should trace back to Freed's work, which is cited as [28] but not referenced here. I recommend including this citation for completeness.

3. The statement on page 6, "This sign can be fixed by anomaly matching: the solution with a minus sign must be chosen if the symmetry group SU(2) suffers from the mod 2 global anomaly discovered by Witten [26] in the UV," requires clarification. Since the expression (2.8) is not real positive for a general X4, the "solution with a minus" does not have a canonical meaning. The branch of the square root should be chosen consistently across the potential four manifolds (with backgrounds) X4. A way to do this is fixing a representative X(0)4 for the bordism class and choosing a branch for the representative. For other X4 in the same bordism class the WZW can be defined using a bordism between X(0)4 and X4. In this particular case one can choose X(0)4 to have the trivial SU(N) background, with which the value in Eq. (2.8) is 1, and then the sign choice for the square root of one corresponds to the SU(2) anomaly.

4. On Page 9: "a topological invariant called the Postnikov class, which is the pair (ˆκL,ˆκR)H3(BSU(N)2;U(1))Z×Z." The meaning of "Hn(BG,U(1))" with G being continuous is ambiguous, as there can be various versions of it. The authors can either comment on this point or avoid it by writing H4(BSU(N),Z) instead.

5. On Page 24: When |Xϕ|1, the Z|Xϕ| subgroup of U(1) does not act on the scalar field ϕ, and thus the discrete subgroup acts on S2 trivially. The IR theory is not S2/Z|Xϕ|-target sigma model if it means the geometric quotient, and it is rather a S2-target sigma model coupled with Z|Xϕ| topological gauge theory. Equivalently, the IR theory can be obtained by gauging Z|Xϕ| subgroup of the U(1) one-form symmetry of S2-target sigma model.

Recommendation

Ask for minor revision

---

## Round 1 · Referee Report · Anonymous (Referee 2) · 2024-10-31

Report

The paper discusses a phenomenon where the quantized level of a Wess-Zumino-Witten coupling term of an infrared pion theory matches the structure constant (Postnikov class) of a 2-group symmetry in the ultraviolet. This is to be contrasted with the more famliar case where the level of the WZW term matches the UV chiral anomaly of an ordinary flavor symmetry. A prominent example discussed is the model with target space SU(N)×S2.

Such a 2-group symmetry structure constrains possible UV completions of the model, as it requires the UV theory to have an exact 1-form symmetry. For instance, it excludes QCD-like completions with non-abelian gauge groups.

The paper is well-written and emphasizes an interesting phenomenon, and it is recommended that the paper is published in SciPost.

Below are minor comments for authors’ consideration:

- On page 2, regarding Ref. [11], to be more precise the mixed anomaly in the pure SU(N) Yang-Mills theory at θ=π is nontrivial only for even N. For odd N, no such mixed anomaly exists, but instead there is the anomaly in the space of θ-angles.

- Around Ref. [12], there are also other early works on higher-group symmetries in (topological) quantum field theories, such as the one by Kapustin and Thorngren https://arxiv.org/pdf/1309.4721.

- In the discussion around Eq. (2.8), it is not so clear why the sign ambiguity should be matched with Witten’s SU(2) anomaly. Perhaps this point could be elaborated more.

Recommendation

Publish (easily meets expectations and criteria for this Journal; among top 50%)

---

## Editorial Decision

resubmitted